# Uncertainties in the Finite Time Lyapunov Exponent in an ocean ensemble prediction model

Mateusz Matuszak[1], Johannes Röhrs[1], Pål Erik Isachsen[12], and Martina Idžanović[1]

[1]Norwegian Meteorological Institute, Henrik Mohns Plass 1, 0371 Oslo, Norway

[2]Department of Geosciences, University of Oslo, P.O. Box 1022, Blindern, 0315 Oslo, Norway

**Correspondence:** Mateusz Matuszak (mateuszm@met.no)

**Abstract.** Lagrangian coherent structures (LCS) are transient features in the ocean circulation that describe particle transport, revealing information about transport barriers and accumulation or dispersion regions. The method of Finite-Time Lyapunov exponents (FTLE) uses Lagrangian data to approximate LCSs under certain conditions. In this study FTLEs are used to characterize flow field features in a high-resolution regional ocean forecast system. Generally, trajectory simulations, such as Lagrangian trajectories, inherit uncertainty from the underlying ocean model, bearing substantial uncertainties as a result of chaotic and turbulent flow fields. As the FTLE characterizes the flow which may impact particle transport, we aim to investigate the uncertainty of FTLE fields at any given time using an ensemble prediction system (EPS) to propagate velocity field uncertainty into the FTLE analysis. In addition, velocity fields often evolve rapidly in time, and we therefore also evaluate the time-variability of FTLE fields. We find that averaging over ensemble members can reveal robust FTLE ridges, i.e. FTLE ridges that exist across ensemble realizations. Likewise, time-averaging can reveal persistent FTLE ridges, i.e. ridges that occur over extended periods of time. In addition, large-scale FTLE ridges are more robust and persistent than small-scale ridges. Averaging of FTLE fields is thus effective at removing short-lived and unpredictable structures, and may provide the means to employ FTLE analysis in forecasting applications that require the ability to separate uncertain from certain flow features.

## 1 Introduction

Ocean currents transport and disperse various environmental tracers, such as nutrients, plankton and pollution. Studying and predicting such transport is of interest and importance for environmental management, especially in the coastal zone. Prediction typically relies on the use of Oceanic General Circulation Models (OGCMs), in which the nonlinear governing equations of motion are first integrated numerically to determine a velocity field. This field is then used to calculate transport and spreading of (synthetic) tracers or particles. In many applications, the aim is not necessarily an exact tracking of individual particles as much as the identification of regions of high or low particle concentration, as well as flow features that may act as dynamical barriers between such regions. To this end, the concept of Lagrangian Coherent Structures (LCS) has received increased attention from the oceanographic community. As the name suggests, LCSs are coherently evolving features in unsteady and chaotic flow fields that can systematically influence particle trajectories (Haller and Yuan, 2000; Tang et al., 2010; Farazmand and Haller, 2012; Haller, 2015). More specifically, LCSs describe coherent morphological features of the flow field that cause

accumulation, spreading and deformation, and identify transport barriers. LCSs have therefore found applications in both process studies and emergency responses, e.g. man-over-board scenarios and oil-spill clean-ups (e.g. Haller and Yuan, 2000; Lekien et al., 2005; Olascoaga and Haller, 2012; Peacock and Haller, 2013; Dong et al., 2021).

Various methods have been proposed to detect LCSs in observed or modeled velocity fields (e.g. d'Ovidio et al., 2004; Shadden et al., 2005; Haller, 2011; Duran et al., 2018; Serra et al., 2020). Among those, the Finite-Time Lyapunov Exponent (FTLE) presents an approximation of LCS that is objective and straightforward to apply (Hadjighasem et al., 2017), and is capable of highlighting areas of particle accumulation or spreading—depending on whether it is computed forward or backward in time—for the spatial and temporal scales on which coastal ocean surface flow varies (e.g. Giudici et al., 2021; Ghosh et al., 2021; Lou et al., 2022). More specifically, the FTLE describes the stretching that a fluid parcel at a given location experiences over a finite time due to the spatially and temporally-varying velocity field. Elongated patches of elevated FTLE values— hereafter referred to as FTLE ridges—may be interpreted as boundaries between coherent structures (flow features identifiable due to their longevity compared to other nearby flows), that is, boundaries between flow features such as eddies, vortices or meandering jets; and it is near such boundaries that a fluid parcel's motion will change drastically (Hussain, 1983; Samelson, 2013; Balasuriya et al., 2016). In unsteady flows, FTLE ridges define time-varying regions exhibiting either an attraction to or repulsion from hyperbolic trajectories (Shadden et al., 2005; Lee et al., 2007; Brunton and Rowley, 2010; Balasuriya, 2012; Balasuriya et al., 2016; van Sebille et al., 2018; Krishna et al., 2023). Under certain conditions, these FTLE ridges may reveal LCSs (Farazmand and Haller, 2012) and provide a diagnostic tool for describing fluid flows that is pertinent to applications of particle transport.

Using FTLE analysis as a detection for LCS has its limitations. For example, a sheared current is not an LCS but will result in high FTLE values, or the detected FTLE ridge may be far away from a true LCS (see example 3 and 4 in Haller (2011)). While more complete methods of LCS detection exist (e.g. Farazmand and Haller, 2012), our study focuses on the FTLE analysis since it provides a straightforward gridded spatial description of Lagrangian transport characteristics that can be analyzed using elementary statistical methods. Given its ease of implementation, we ultimately aim to examine the potential use of FTLE analysis as a practical tool for applications in operational oceanography, e.g. oil-spill modeling.

More specifically, the current study will examine the usefulness of an FTLE approach to transport and dispersion modeling in light of the *uncertain* nature of any ocean model forecast. Due to the nonlinear and highly chaotic nature of real ocean flows—as well as the flow in high-resolution ocean models—small errors in the knowledge or specification of the velocity field may yield large perturbations in estimated particle trajectories. Furthermore, even with a perfect knowledge of the velocity field, uncertainties in particle's initial position or time of release may grow into large uncertainties over time. Thus, despite the potential usefulness of LCS or FTLE analysis outlined above, the need to address the uncertainty and errors in e.g. the underlying current velocity remains. Can the uncertainty in forecasted FTLE fields be quantified? A common way to address prediction uncertainties in geophysical flow fields is to use Ensemble Prediction Systems (EPS). Instead of issuing a single deterministic integration of the circulation model, an ensemble of model realizations is obtained by time-integrating the model with variations in the initial conditions and boundary conditions (e.g. the atmospheric forcing). The ensemble is hence intended

to span out the possible states of the system (e.g. Lebreton et al., 2012; Idžanović et al., 2023). While common in weather prediction, this method is in its infancy in regional ocean prediction (Thoppil et al., 2021).

The impact of general flow variability on FTLE and other LCS analyses has received some attention, specifically related to the question of whether time-persistent features can be identified in a nonlinear and chaotic flow field (e.g. Olascoaga et al., 2006; Gouveia et al., 2020; Dong et al., 2021). Some studies have also been conducted with the aim of addressing the uncertainty aspect, using ocean EPSs. (e.g. Wei et al., 2013; Guo et al., 2016; Wei et al., 2016; Balasuriya, 2020; Zimmermann et al., 2024). Here, we wish to further elaborate on how FTLE analysis can give information on coherent flow structures, despite the presence of time variability and uncertainty in the forecast. We specifically distinguish between *persistence* and *robustness* of flow features: We refer to persistence in relation to flow features that remain at their location over an extended period of time, hence provide usefulness for applications that use an analysis and assume the flow field remains in a similar state. Then we refer to robustness in a prediction of flow features if a majority of a model's ensemble members indicate a similar outcome such that the forecasted FTLE have a high probability to be realized in nature.

Our study region will be the continental shelf, continental slope and deep ocean basin off Lofoten-Vesterålen in Northern Norway, a region of considerable importance for both the marine climate and marine ecosystem in the northern North Atlantic. In section 2, we describe the operational EPS ocean forecasting system for this region and provide an outline of how the FTLE analysis is performed. In section 3, we present results invoking time averages and ensemble averages of FTLE fields, respectively. In section 4, we draw conclusions on temporal and seasonal variability of FTLE and uncertainties in the FTLE analysis. Finally, we discuss implications on the applicability of the FTLE analysis in uncertain flow fields as a tool in operational forecasting.

## 2 Data and methods

### 2.1 Study region

The bathymetry and modeled surface currents around the Lofoten-Vesterålen (LoVe) archipelago along northern Norway's coast are shown in Figure 1. The continental shelf sea off LoVe is known to be a hot spot for fisheries due to its high concentrations of nutrients, which form feeding grounds and spawning banks for marine life (Sundby and Bratland, 1987; Sundby et al., 2013). Transport of relevant nutrients has been widely studied (e.g. Adlandsvik and Sundby, 1994; Röhrs et al., 2014), and the Finite-Size Lyapunov Exponent (FSLE) analysis presented in Dong et al. (2021) sheds light on possible mechanisms for cross-slope transport of nutrients that could play a role in sustaining biological production.

The LoVe region is characterized by complex bottom topography and a steep continental slope that steer the region's primary large-scale currents (Sundby, 1984), namely the Norwegian Atlantic Slope Current (NwASC) (Rossby et al., 2009) and the Norwegian Coastal Current (NCC) (Gascard et al., 2004). The complex coastline and the Vestfjorden embayment directly guide the path of NCC, causing complex flow features, including strong tidal currents through Moskstraumen—one of the many straits that cut through the archipelago (Børve et al., 2021). During winter, southerly winds enhance the onshore Ekman

transport and water mass accumulation along the coast, thus accelerating large-scale currents after geostrophic adjustment (Mitchelson-Jacob and Sundby, 2001).

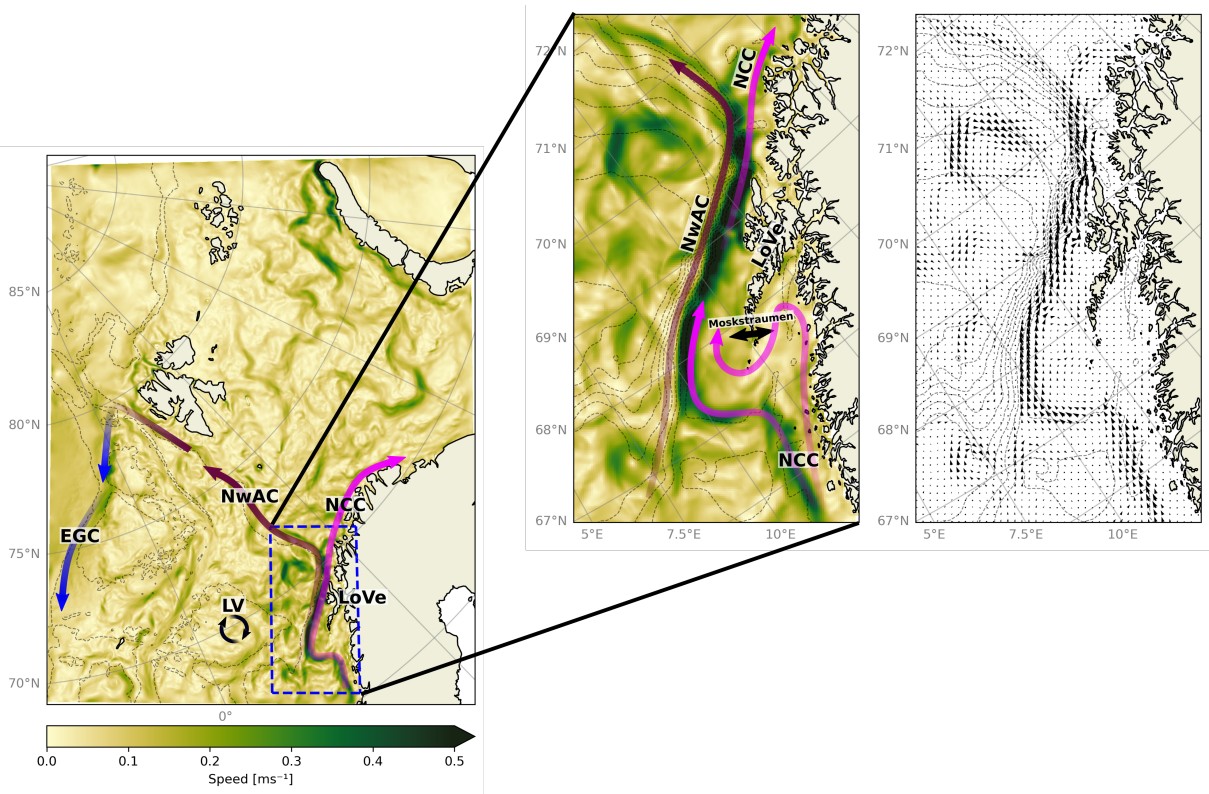

**Figure 1.** Average ocean current speed for the period 2023-02-01 to 2023-02-28 over the full Barents-2.5 model domain (left panel). Pink arrows indicate the Norwegian Coastal Current (NCC). Brown arrows indicate the Norwegian Atlantic Current (NwAC). Blue arrows indicate the East Greenland Current (EGC). The circular black arrows indicate the Lofoten Vortex (LV). Dashed blue lines highlight the region of interest for this study, which is enlarged in the middle and right panels (speed and current directions, respectively). The two-headed black arrow in the middle panel indicates Mokstraumen. Bathymetric contours are indicated by gray dashed lines.

The NwASC and NCC meet right off the LoVe archipelago. The steep continental slope, combined with a narrow shelf, sets up steep fronts that host a range of flow instabilities. The result is the most intense mesoscale eddy field in all of the Nordic Seas and vigorous exchanges of heat, salt and nutrients between the shelf and deep ocean (Koszalka et al., 2013; Isachsen, 2015; Trodahl and Isachsen, 2018). As such, the region offers a particular challenge with respect to accurate modeling of currents and transport.

## 2.2 Regional Ocean Ensemble Prediction System

We use flow data from Barents-2.5 EPS (Röhrs et al., 2023), an ensemble prediction system based on the Regional Ocean Modeling System (ROMS; Shchepetkin and McWilliams, 2005). The model has a 2.5 km horizontal grid size and hourly

temporal resolution, and covers the Barents Sea, the coast off northern Norway and Svalbard (see Figure 1). The EPS consists of 24 members, divided into four sets of six members. The sets are initiated with a 6-hour delay, at 00 UTC, 06 UTC, 12 UTC, and 18 UTC, with a forecast period of 66 hours. Each member is initialized by its own state from the previous day in order to preserve sufficient spread in the ensemble and the members are run independently of each other. The ensemble spread is
further controlled by the Ensemble Kalman Filter data assimilation scheme, which controls the spread of observed variables (Evensen, 1994; Röhrs et al., 2023). The first member in each set (four members) is forced by the most recent atmospheric conditions from AROME-Arctic (Müller et al., 2017). The remaining members are forced by 20 members randomly drawn from the integrated forecast system developed by the European Centre for Medium Range Weather Forecasts (ECMWF-ENS) (Röhrs et al., 2023).

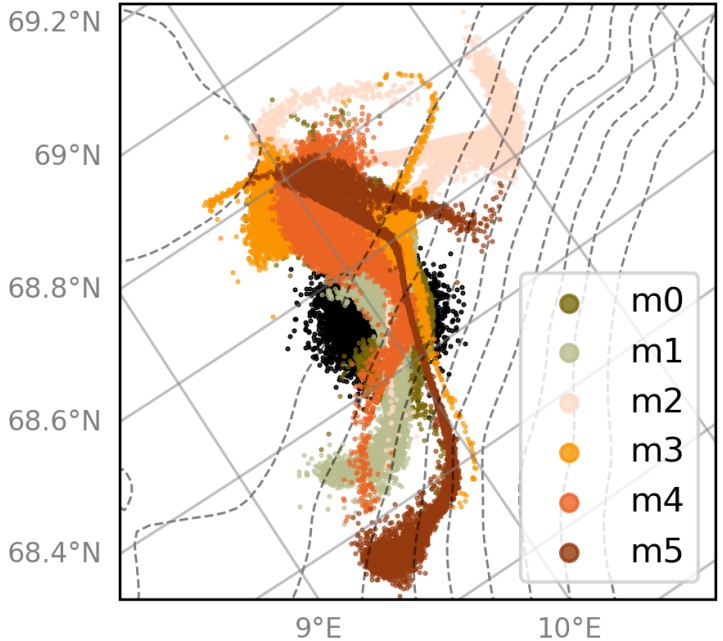

**Figure 2.** A particle cluster advected for 48 hours from 2023-02-01 using velocity fields from six Barents-2.5 EPS members. Black dots mark the particle clusters initial position. Bathymetric contours are indicated by gray dashed lines.

A detailed analysis of particle transport in Barents-2.5 EPS is discussed in de Aguiar et al. (2023), but Figure 2 exemplifies the effect of flow field uncertainty on clusters of particles that have been advected using velocity fields from different ensemble members. We see that after 48 hours the particle clusters have taken a distinct shape based on the velocity field of the used ensemble member, and an estimated trajectory uncertainty can be obtained from the spread. The trajectory uncertainty is small when flow velocities are similar across the ensemble and increases when there is a large discrepancy between them.

## 2.3 The Finite-Time Lyapunov Exponent

A particle will be advected in the presence of an underlying flow field. The trajectory may be obtained by integrating along the encountered flow field:

$$\boldsymbol{x}\left(t\right) = \boldsymbol{x}_0 + \int_{t_0}^{t} \boldsymbol{u}\left(\boldsymbol{x}\left(\tau\right), t\right) \mathrm{d}\tau. \tag{1}$$

Here, $\boldsymbol{x}\left(t\right)$ is the position of a particle at time $t$ advected from its initial position $\boldsymbol{x}_0$ using the time-variable velocity field $\boldsymbol{u}$ along the evolving trajectory locations $\boldsymbol{x}\left(\tau\right)$. In this study, particle trajectories are calculated by OpenDrift (Dagestad et al., 2018), an open-source Python based software for Lagrangian particle modeling developed at the Norwegian Meteorological Institute.

The Lyapunov exponent is a parameter which describes the separation rate between two neighboring particles in a chaotic system. The focus is on exponential-in-time separation, for which the distance $\delta_t$ between the two particles at time $t$ is

$$\delta_t \approx \delta_0 e^{\sigma t}, \tag{2}$$

where $\delta_0$ is the initial separation and $\sigma$ is the Lyapunov exponent, i.e. the separation rate (Rosenstein et al., 1993). Mapping out estimates of the Lyapunov exponent, based on the observed trajectories of a large number of particles over finite time-intervals, allows one to search for patches, or 'ridges', of particularly high separation rates (Pierrehumbert and Yang, 1993).

Fine-Time Lyapunov Exponents are calculated from flow fields provided by an OGCM following the method described in Haller (2001), Shadden et al. (2005) and Farazmand and Haller (2012). The two-dimensional (2D) movement of fluid parcels from their initial positions $\boldsymbol{x}_0 = (x_0, y_0)$ at time $t_0$ to their final positions at time $t$ is described by a flow map $\boldsymbol{F}_{t_0}^{t}\left(\boldsymbol{x}_0\right)$. As multiple fluid parcels are transported by the flow, the distance between neighboring fluid parcels is likely to contract or expand over the time interval. At each point in space, the change in separation between fluid parcels can be described by the Jacobian of $\boldsymbol{F}_{t_0}^{t}\left(\boldsymbol{x}_0\right)$:

$$\boldsymbol{\nabla}\boldsymbol{F}_{t_0}^{t}\left(\boldsymbol{x}_0\right) = \begin{bmatrix} \frac{\partial x}{\partial x_0} & \frac{\partial x}{\partial y_0} \\ \frac{\partial y}{\partial x_0} & \frac{\partial y}{\partial y_0} \end{bmatrix}, \tag{3}$$

where $(x, y)$ is the final position of a fluid parcel which was initially located at $(x_0, y_0)$. These positions may be obtained from Eq. 1. The matrix entries in Eq. 3 are the partial derivatives of the final position relative to their initial position. Eq. 3 is used to define the Cauchy-Green strain tensor $\boldsymbol{C}_{t_0}^{t}\left(\boldsymbol{x}_0\right)$ (Truesdell and Noll, 2004), which describes the deformation in the system

$$\boldsymbol{C}_{t_0}^{t}\left(\boldsymbol{x}_0\right) = \left[\boldsymbol{\nabla}\boldsymbol{F}_{t_0}^{t}\left(\boldsymbol{x}_0\right)\right]^{*} \boldsymbol{\nabla}\boldsymbol{F}_{t_0}^{t}\left(\boldsymbol{x}_0\right). \tag{4}$$

The FTLE is then defined using $C_{t_0}^{t}$:

$$\sigma_{t_0}^{t}\left(\boldsymbol{x}\right) = \frac{1}{|T|} \ln \sqrt{\lambda_{max}\left(\boldsymbol{C}_{t_0}^{t}\right)}, \tag{5}$$

where $T$ is the time interval over which the FTLE is computed and $\lambda_{max}\left(\boldsymbol{C}_{t_0}^t\right)$ is the largest eigenvalue of $\boldsymbol{C}_{t_0}^t\left(\boldsymbol{x}_0\right)$ corresponding to the dominant stretching direction (eigenvector) in the system. For simplicity, subscripts $t_0$ and $t$ are hereafter dropped. If one uses FTLE as an LCS detection tool, a forward-in-time computation will correspond to repelling LCS, whereas
a backward-in-time computation will correspond to attracting LCS (Haller, 2001; Shadden et al., 2005; Farazmand and Haller, 2012).

In this study, we investigate the FTLE computed from backward-in-time integrations. Furthermore, the study is motivated by typical uses of ocean forecasting models, which are decision-support tools for search-and-rescue operations, oil spill, iceberg forecasts, and similar trajectory analyses. These often operate at time-scales from a few hours up to a few days, and therefore,
we predominantly use $T = 24$ hours for the FTLE computations. We also provide some discussion of the choice of $T$ in Sec. 3.1

Variations among ensemble members and over time are expected due to perturbed and time-evolving velocity fields. FTLE averages over ensemble members and over time periods will thus be calculated to characterize robustness and persistence, respectively. For each such analysis, we first compute the FTLE fields from a set of flow fields and thereafter calculate averages
over those FTLE fields, which is similar to the D-FTLE mean method discussed in Guo et al. (2016). We define the ensemble and time averages as:

$$\overline{F}_m = \frac{1}{N} \sum_{i=1}^{N} \sigma\left(m_i\right) \tag{6}$$

$$\overline{F}_t = \frac{1}{M} \sum_{j=1}^{M} \sigma\left(\tau_j\right), \tag{7}$$

where $\sigma\left(m_i\right)$ represents the FTLE field for ensemble member $m_i$ and $\sigma\left(\tau_j\right)$ represents the FTLE field over a specific time
period $\tau_j$. For example, for time interval $T = 24$ hours, $\tau_j$ refers to the specific daily FTLE field selected from a series of multiple daily fields. $N$ is the total number of distinct FTLE fields obtained from different ensemble members (in this study, $N$ will generally equal 24, which is the total number of Barents-2.5 EPS members), while $M$ denotes the number of FTLE fields obtained from distinct time periods.

It is expected that averaging FTLE fields will smooth out non-robust and non-persistent features while highlighting robust
and persistent features, ultimately indicating regions where high FTLE values are statistically likely to form at any one time or frequently form over time. The presence of coherent 'ridges' in the averaged FTLE fields are thus potential candidates for robust or persistent material accumulation regions and, possibly, indications of transport barriers.

A way to try to identify features that are *both* robust and persistent may be to first ensemble average the velocity field at each time, then calculate the FTLE field from the averaged field, and finally do time-averaging. Presumably, this will remove
uncertain flow features right from the start, and the resulting FTLE field may prove to be more persistent. Such a procedure may have its own inherent problems, as ensemble averaging velocity fields may also produce unrealistic flow features, e.g. strange flow structures will emerge if the members predict coherent vortices which are slightly perturbed in location from

member to member. We nevertheless explore this approach as part of our examination of the variability of FTLEs in the present high-resolution ocean EPS.

## 3  Results

We first have a quick look at how the choice of integration time impacts the FTLE field, assessing this in relation to applications relevant for operational oceanography. We then look at whether there is in fact any persistence or robustness in FTLE fields over the dynamically active LoVe region. Finally, we do a spectral analysis of the FTLE field in the attempt to pin-point the resolution needed for practical use in an operational forecasting system.

### 3.1  Integration time

Backward-in-time FTLE fields, all based on velocities from the first ensemble member and all starting from the same $t_0$ at 2021-12-31 but using different integration lengths $T$, are shown in Figure 3. The values are normalized to be between $0$ and $1$ using the following equation:

$$\widetilde{\sigma}\left(x,y\right) = \frac{\sigma\left(x,y\right) - \min\left(\sigma\right)}{\max\left(\sigma\right) - \min\left(\sigma\right)}, \tag{8}$$

as the FTLE values tend to decrease with increasing $T$. Here, $\widetilde{\sigma}_i$ is the normalized FTLE value at position $(x,y)$ and $\min\left(\sigma\right)$ and $\max\left(\sigma\right)$ are the minimum and maximum FTLE values in the domain. As noted by e.g. Wilde et al. (2018) and Peng and Dabiri (2008), a longer integration time tends to result in sharper FTLE ridges. We note, however, that the overall structure of the FTLE field is not overly sensitive to the integration period within the integration length of 12 to 72 hours, although the features in the field are more detailed for the 72-hour period than for 12 hours. For even longer integration periods there is clear indication that distinct FTLE ridges in the energetic flow regions over the continental shelf and slope are smeared out. A plausible interpretation is that the ability of the FTLE field to describe flow field features depends on the integration period that matches the time scale of the dynamics. This advective time, which scales as $L/U$ (for velocity scale $U$ and length scale $L$), then depends on environmental conditions. So, capturing highly energetic small-scale features associated with the fronts over the continental shelf and slope requires short integration times. In contrast, low-energy and large-scale features over the deep basin are slow enough to be well represented by FTLE integrations that have been conducted over several days.

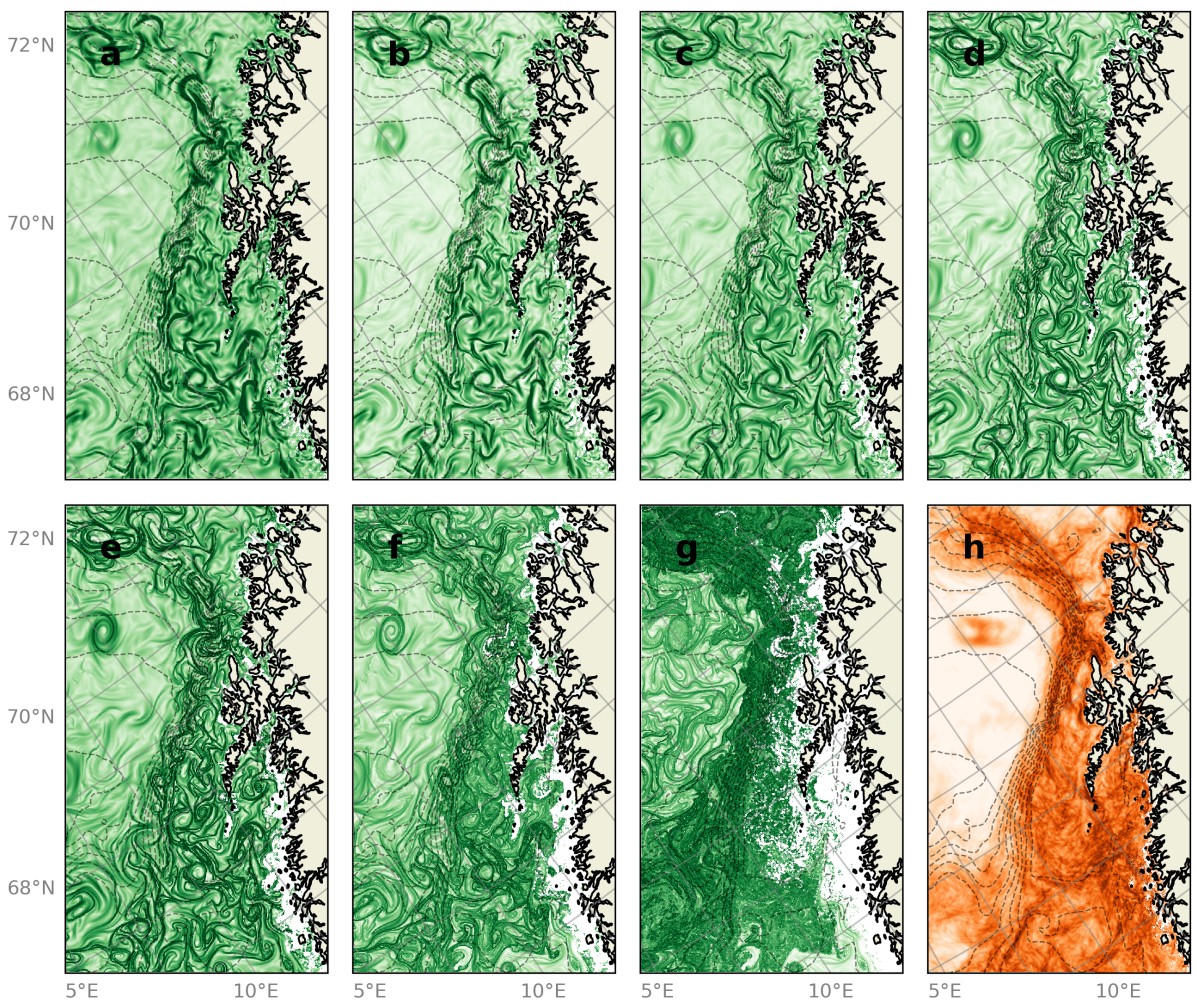

**Figure 3.** Normalized FTLE computed from the first ensemble member and all starting at 2021-12-31 but using different time windows $T$ : a) 6 hours, b) 12 hours, c) 24 hours, d) 48 hours, e) 72 hours, f) 168 hours (7 days) and g) 672 hours (28 days). h) Monthly average of FTLE fields computed with $T = 24$ hours. Bathymetric contours are indicated with gray dashed lines.

A time-averaged FTLE field is also shown in Figure 3h. This is included to illustrate that the FTLE analysis based on long integration periods (e.g. over 168 and 672 hours) are distinctly different from the time-averaged FTLE field of several 24-hour integration periods. The time-averaged FTLE field should be interpreted as highlighting regions that are *typically* abundant with high FTLE values over the time period. The calculation reveals that in this particular region, structures forming high 200 FTLE values are most often found over the continental slope. In contrast, the FTLE structures appearing over the deep basin when FTLEs are calculated over long integration periods wash out in the average description.

## 3.2 Persistence over time

Velocity magnitude fields and corresponding backwards FTLE fields from the same first member of the Barents-2.5 EPS (henceforth called the reference member) are shown in Figure 4 for three example dates one week apart in January 2023, along with the monthly-averaged velocity and FTLE fields. As expected, the continental slope current is shown to be persistent over the time period. However, the current meanders and its intensity changes over time, and this time variability projects onto the FTLE fields. Thus few FTLE ridges are seen to stay the same between time frames. And yet, there is clearly a concentration of high-magnitude FTLE ridges over the steep continental slope during this time period—as effectively summarized by the time-averaged FTLE field, $\overline{F}_t$. The interpretation is that strong FTLE ridges are expected to be frequent along the continental slope, at least over this sample time period, even though the FTLE average over time does not yield detailed information about how these look like as individual features.

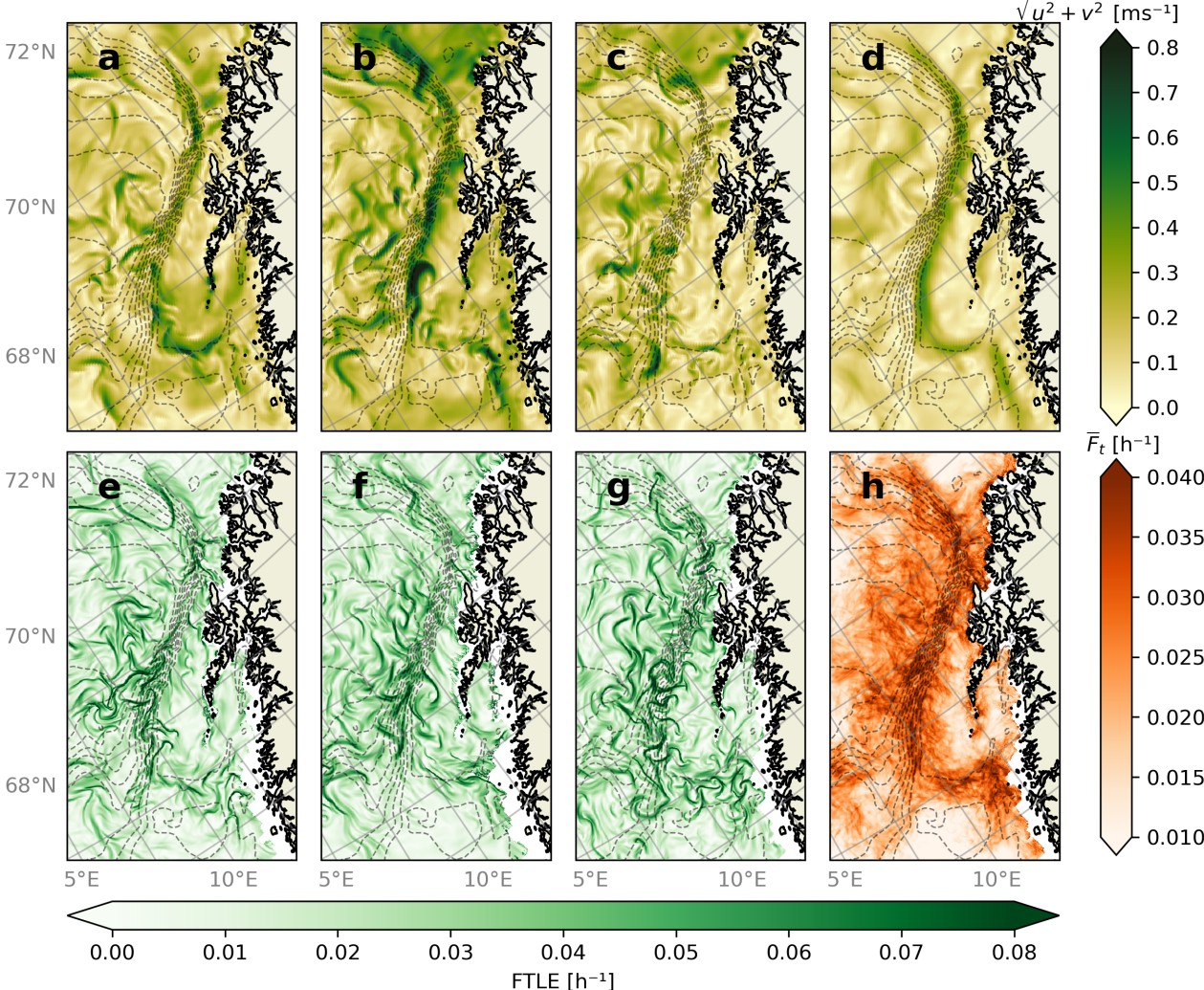

**Figure 4.** Instantaneous velocity fields (top row) from the reference member of Barents-2.5 EPS at a) 2023-01-02, b) 2023-01-08, and c) 2023-01-15 at 00:00. d) Averaged velocity field for January 2023. Backwards FTLE fields (bottom row) computed with $T = 24$ hours over e) 2023-01-02, f) 2023-01-08, and c) 2023-01-15. h) Monthly average of daily FTLE fields for January 2023. Bathymetric contours are indicated with gray dashed lines.

To further highlight the permanent impact of the continental slope, seasonally-averaged velocity fields from the reference member are shown in Figure 5 along with seasonal FTLE averages, $\overline{F}_t$, computed from daily FTLE fields from the summer and winter seasons. The slope current is placed similarly in both seasons, although it is stronger during winter, likely due to a geostrophic adjustment to the sea surface tilt, as discussed in Sec. 2.1. In contrast, $\overline{F}_t$ changes drastically between the two seasons. We see that strong values in the FTLE field along the continental slope occur much more frequently during winter.

Large values can be seen for both seasons near the coastline, which are suspected to be produced by strong horizontal velocity shear near the coastal regions.

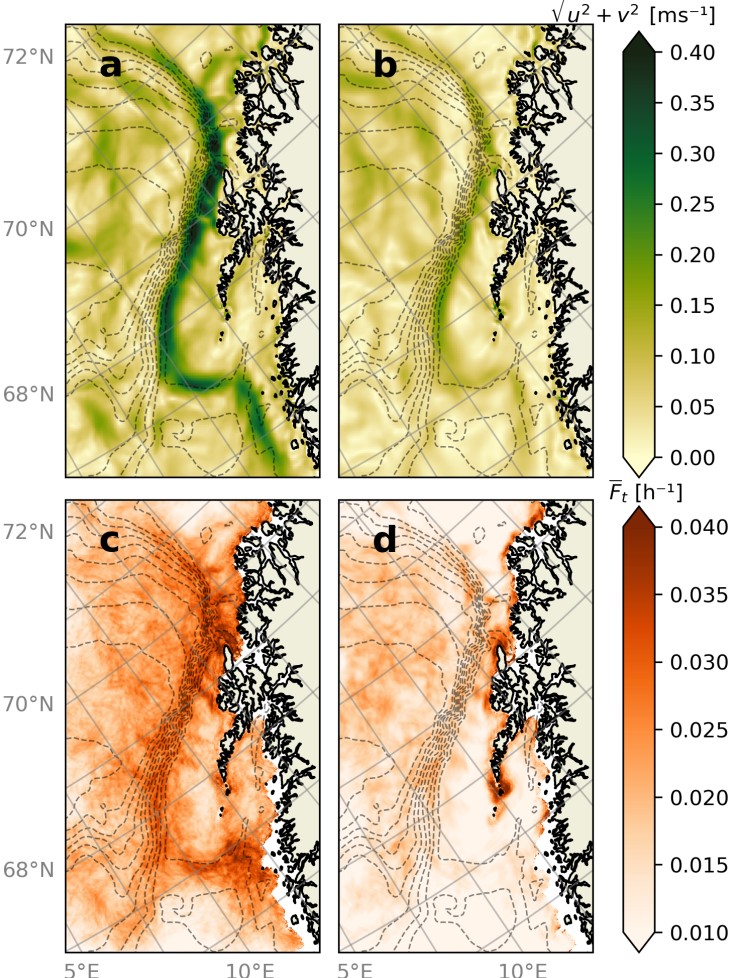

**Figure 5.** Seasonal velocity averages for a) winter and b) summer, and seasonal FTLE averages for c) winter and d) summer. Months included in the winter season are December of 2022, January, and February of 2023. Months included in summer season are June, July, and August of 2023. Bathymetric contours are indicated with gray dashed lines

Near the coast, a region of high $\overline{F}_t$ around Moskstraumen strait (at the southern tip of the LoVe archipelago; see Figure 1), especially during summer, is directly connected to the formation of strong jets at the strait exit. The direction of the current through the strait is dependent on the tidal phase (Børve et al., 2021). After closer investigation using $T = 2$ hours for the FTLE time interval, high values in the FTLE field tend to form only on one side of Moskstraumen at any particular time, depending on the current direction and thus the tidal phase. Therefore, a predictable tidal-dependent periodic variability of FTLEs may exist here. FTLE has previously been shown to be highly sensitive to the tidal phase (Zhong et al., 2022). However, the model's

spatial resolution may be too low to fully resolve the currents in this region, and a closer investigation into the dynamics at play here will need to be left for a future high-resolution model study.

## 3.3 Robustness over ensemble realizations

We turn next to the concept of robustness of FTLE fields, that is the extent to which FTLE fields computed using flow fields from different EPS realizations are similar. As an example, velocity fields and FTLE fields, all calculated for a specific time
but for three randomly selected ensemble members, are shown in Figure 6, along with the ensemble-averaged velocity and FTLE field. We see that the individual members all contain a strong current along the continental slope, which has also been shown to be a time-persistent current. However, and as expected from a highly nonlinear and chaotic flow field, the position and strength of individual eddies and current meanders vary considerably between members. This is certainly the case for small-scale structures along the slope current. But some larger-scale mesoscale structures over the deep ocean, e.g. a vortex
in the south-western corner of the domain, are actually predicted by all three ensemble members. Such large-scale features thus survive the smoothing inherent in the ensemble-averaged velocity field, whereas most individual small-scale structures are washed out. Plainly speaking, the EPS gives a low confidence that any of these small-scale structures actually exist in the real ocean at their specific location at this particular time.

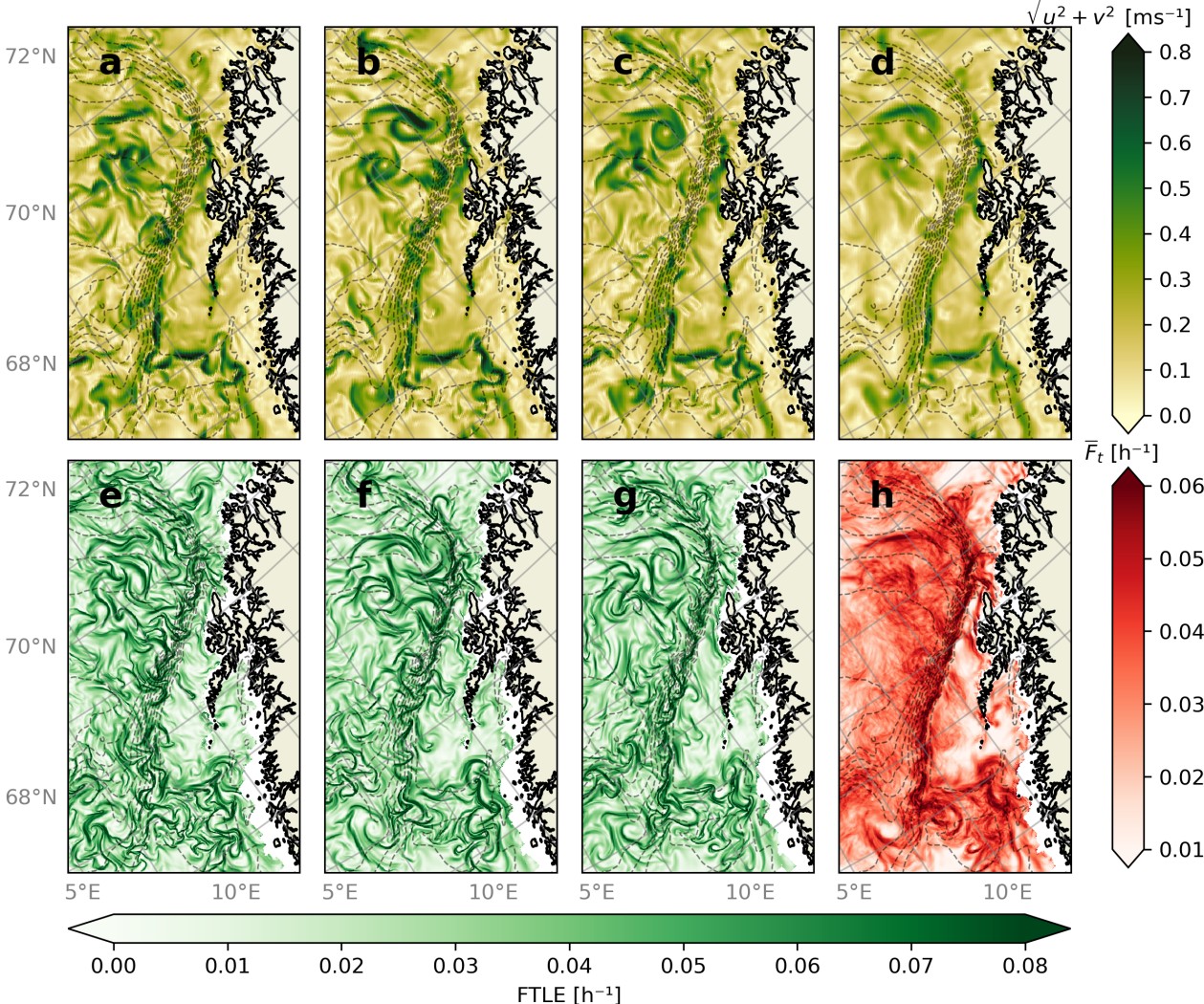

**Figure 6.** a, b and c) Velocity fields from three different Barents-2.5 EPS ensemble members at the LoVe region at 2023-02-02 00:00 and d) ensemble averaged velocity field over all EPS members. e, f and g) Backwards FTLE fields computed over 2023-02-01 for the members shown in a, b and c. h) The ensemble-averaged FTLE field. Bathymetric contours are indicated with gray dashed lines

The flow variability within the ensemble again projects directly onto the FTLE values, and, as expected, there is generally little one-to-one agreement between the three ensemble members displayed here. However, we see that all the members of the ensemble predict high FTLE values along the continental slope, as well as in the eddy-dominated deep basin region around 70.5° N. But the exact position and strength of FTLE maxima vary considerably and even more so than the velocity field itself. Again, we must interpret this as indication that the FTLE field from a single model realization may not reflect conditions in the real ocean at any specific time.

Thus, instead of inspecting the FTLE fields of each member individually, a study of the ensemble-averaged FTLE field, $\overline{F}_m$ (Figure 6h), allows us to detect robust flow features; high values will be present in $\overline{F}_m$ where multiple (but not necessarily all) individual members predict high FTLE levels. In the situation studied here, $\overline{F}_m$ shows a long and continuous feature tangent to the continental slope. However, in individual members the features formed by high FTLE values are seen to be disjointed, thinner and often not tangent to the continental slope. The eddy-dominated region around $70.5°$ N also contains high averaged FTLE values, but these are smoother than over the continental slope, thus presumably reflecting typical occurrences of strong FTLE ridges but also a lower impact of bottom bathymetry. An FTLE average may thus yield both distinguishable features in the domain, which can be considered as robust features, as well as large smooth fields of higher FTLE values, which indicate that FTLE ridges are likely to be found here but are more variable across the ensemble.

In light of the 'noise reduction' resulting from ensemble averaging, it is natural to ask whether the time persistence examined in the previous section is impacted by such ensemble averaging. The order of averaging could be done in several ways but, as previously mentioned, by first ensemble-averaging the velocity field we may remove uncertain flow features right from the start, that is, before we study persistence. In Figure 7 we show monthly $\overline{F}_t$ produced from such ensemble-averaged velocity fields. The calculation is done for three winter months, and the January field (middle panel) can be compared with Figure 4h in which the monthly-averaged FTLE field—from one member only—is shown. Quite clearly, much of the time-averaged FTLE structure in a single member is removed by using ensemble-averaged velocities. We notice, however, that certain FTLE ridges remain well-defined in Figure 7, most notably around the continental slope which steers the mean flow. This is especially true for the February average. Thus, by first ensemble-averaging the velocity field we are able to remove non-robust flow features right from the start and, in turn, highlight what are more likely to be actual persistent FTLEs.

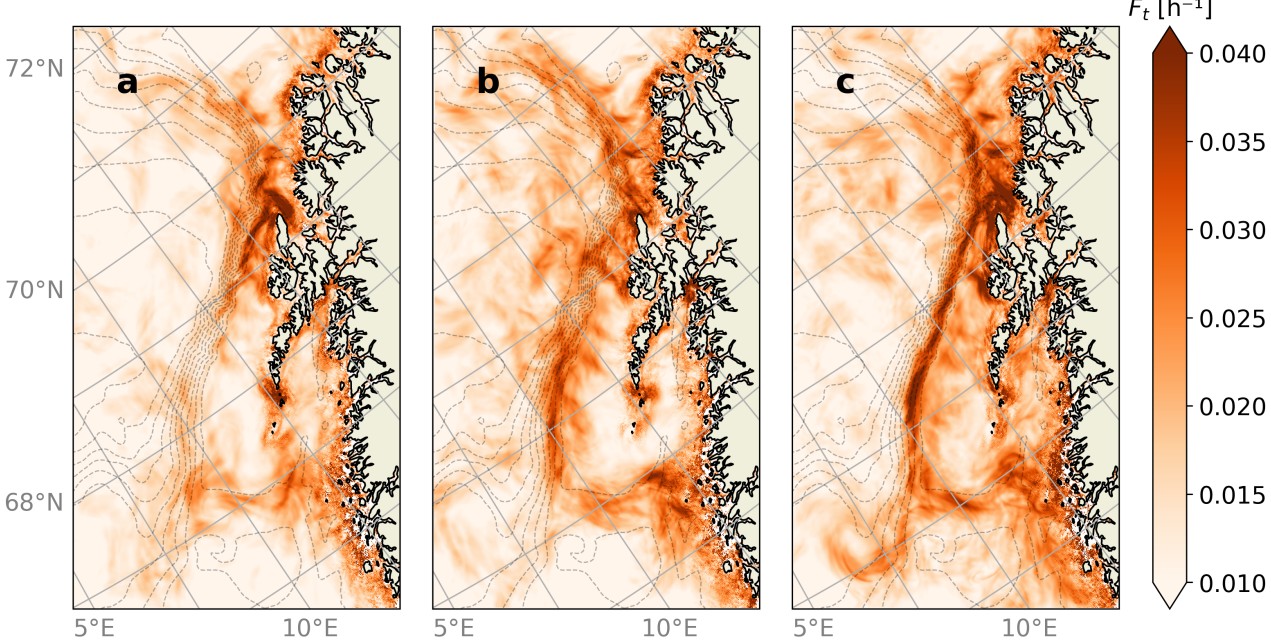

**Figure 7.** Monthly time averaged FTLE fields produced from ensemble-averaged velocity fields for a) December, b) January, and c) February. Bathymetric contours are indicated with gray dashed lines.

## 3.4 Impacts of ensemble and time averaging on FTLE variance

We wish to systematically examine how ensemble or time averaging impacts the FTLE spatial variance—as well as the spectral distribution of this variance. First, the spatial variance of the FTLE field was computed over a 200 km×600 km region away from the coast, and this was done after averaging over an increasing number of members or days. The results are shown in the top two panels of Figure 8. As expected, the calculations show that FTLE variance decreases as more members or time frames are considered in the averages. Simply put, the averaging acts to smooth the field. The results also suggest that the smoothing

rate of FTLE fields is independent of season. However, the absolute level of the variance is lower during summer as the FTLE values themselves are generally lower compared to winter (see also Figure 5).

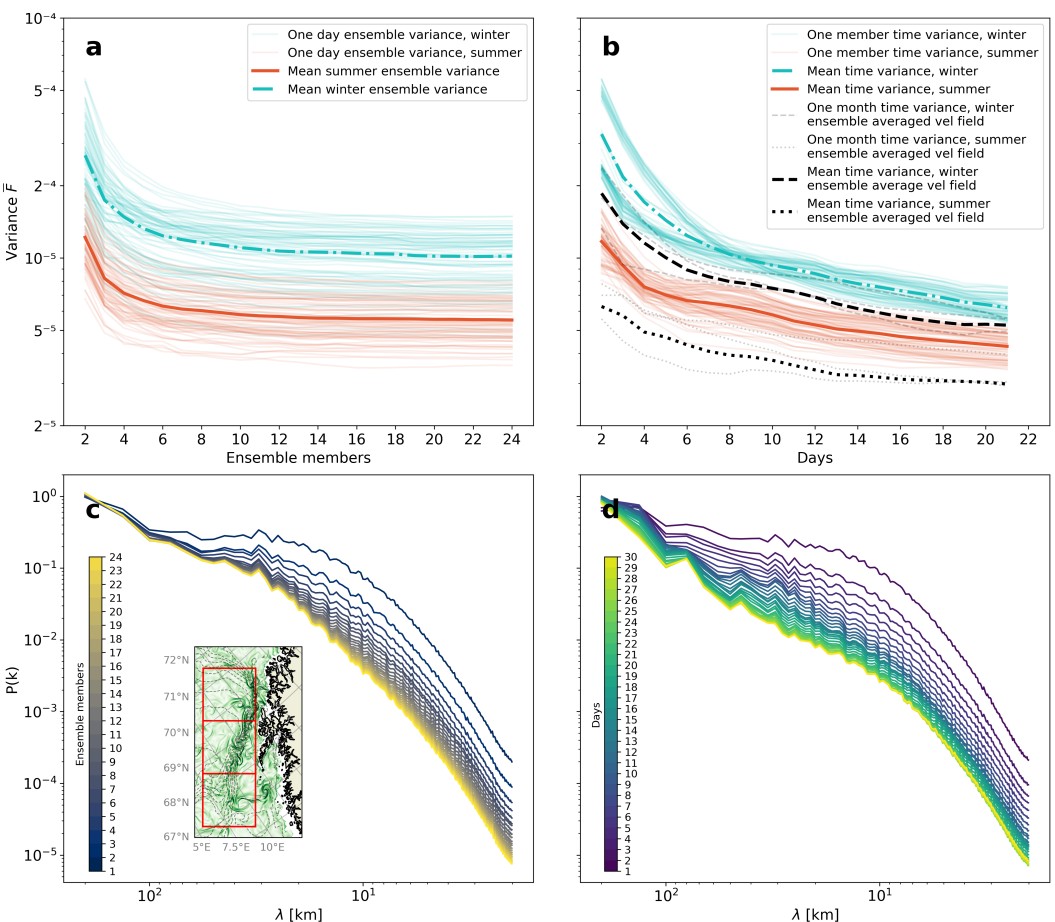

**Figure 8.** Spatial and spectral variance of FTLEs and averaged FTLEs. a) Spatial variance over the ensemble average. Thin blue and red lines indicate individual days during the winter and summer seasons, and the thicker blue and red lines are the average of the thin lines. b) Spatial variance over the time average. Thin blue and red lines indicate the evolution of variance during the winter and summer months as up to 21 days are considered in the time average for each member, with the thick blue and red lines showing the average of the thin lines. The dashed and dotted black lines show the evolution of variance after computing the FTLE field from ensemble averaged velocity fields. c) Spectral distribution of spatial FTLE variance as a function of wavelength (in kilometers) in ensemble averages as an increasing number of members are considered in the average, averaged over all days in January 2023. The red squares in the randomly selected FTLE field in c) indicate the regions selected for the spatial and spectral variance computation. d) Spectral distribution of spatial FTLE variance for time averages as an increasing number of days are considered in the average, starting from 2023-01-01, averaged over all members. Colorbars in c) and d) indicate how many members or days the FTLE fields are averaged over, respectively.

The strong decay at the beginning of the $\overline{F}_m$ variance (Figure 8a) may be taken as an indication that there is a good level of spread in the ocean model ensemble. The spatial variance of $\overline{F}_m$ then tends to stabilize once $\sim 10$ members are considered,

revealing the existence of more predictable flow features in the ensemble. As such, the transition towards flattening may be taken as an indication of the number of independent members in the EPS.

In contrast, the spatial variance of $\overline{F}_t$ (Figure 8b) continues to drop as the averaging period lengthens, most likely reflecting that there is real variability in ocean flows at all time scales. Finally, we see that the spatial variance over time of the FTLE fields computed from ensemble-averaged velocities (hereafter $var_{ens}$) is lower than the $\overline{F}_t$ variance. This makes sense, as the most unpredictable and chaotic flow features have previously been smoothed due to velocity averaging. However, $var_{ens}$ also does not stabilize over the time period, again reflecting that variability in the ocean is continuously spread over all time scales.

Then we examine how averaging impacts the distribution of variance over different spatial scales. For this we look at 2D spectra, using the discrete cosine transform (DCT), as proposed by Denis et al. (2002). Again, we select the 600 km×200 km region away from landmasses used for the spatial variance and split it into three 200 km×200 km non-overlapping boxes. The DCT produces an $N_i$ by $N_j$ field $F(m,n)$ of spectral coefficients, where $m$ and $n$ are non-dimensional wavenumbers. For a square domain where $N_i = N_j = N$, the wavelength is given by

$$\lambda = \frac{2N\Delta}{k},\tag{9}$$

where $\Delta$ is the grid spacing and $k = \sqrt{m^2 + n^2}$ is a radial wavenumber.

Spectra, showing variance density as a function of wavelength, are shown in the two lower panels of Figure 8. The spectra have been computed over each of the three boxes and then averaged. The variance density drops for smaller scales, in line with the general tendency for geophysical spectra to be red-shifted. In line with the total FTLE variance studied above, spectral levels drop as more members and time frames are included in the averages. Spectral levels diminish similarly for shorter wavelengths but significantly faster for $\overline{F}_t$ (Figure 8c) than for $\overline{F}_m$ (Figure 8d) for long to mid-sized wavelengths.

As seen above, the decay of variance as more FTLE fields are included in an average behaves differently for ensemble and time averaging (Figure 8a vs. Figure 8b). In other words, some FTLE ridges are robust when averaged over many ensemble members but do not achieve persistence as the temporal averaging window extends, at least over weekly periods relevant for operational forecasting. As the spectra reveal, the stronger decline at the beginning of the averaging is due to small-scale FTLE ridges experiencing a strong and quick smoothing due to averaging. The slow-down for more averaging elements reflects that mostly large-scale FTLE ridges are left. Robustness is particularly prominent at the larger spatial scales (Figure 8c), where little decay is noticed as more than 2-5 ensemble members are included in the average for long wavelengths. The spatial variance of $\overline{F}_m$ stabilizes as the large scale features that are left in the system are highly robust. $\overline{F}_t$, in contrast, continues being smoothed at all scales due the formation, drift, deformation and dissipation of FTLE ridges happening at all scales, albeit slower at the largest scales.

## 4 Summary and discussion

Features in ocean surface circulation in a coastal region off northern Norway, as described by FTLE analysis, have been investigated in terms of their *persistence* in time and *robustness* across ensemble members in a high-resolution ocean EPS. The

basic question is to what extent FTLE calculations are actually useful in operational forecasting given the chaotic and time-variable nature of ocean flows. Time and ensemble averages have therefore been computed as an attempt to identify robust and persistent FTLE ridges, respectively, while averaging out transient and uncertain features. Below, we summarize and discuss some of the key findings.

## 4.1 FTLE as an indicator of LCS and transport barriers

LCSs describe attracting and repelling properties of fluid flows, as well as define transport barriers (Haller and Yuan, 2000; Farazmand and Haller, 2012; Haller, 2015). FTLE ridges have been discussed as possibly indicating the presence of LCSs, however with clear limitations. For instance, horizontal velocity shear may produce large FTLE values but will not yield normal attraction towards or normal separation from the FTLE ridge, which is what characterizes LCS. Thus, such FTLE ridges are not indicative of LCSs (Haller, 2002; Branicki and Wiggins, 2010; Haller and Sapsis, 2011; Karrasch and Haller, 2013). Other limitations exist, but a strength of the FTLE approach is that it allows for simple statistical analysis of flow field features that in some instances point to the existence of LCSs.

At the very least, the systematic patterns in the FTLE fields over the continental slope off LoVe do suggest that the method picks up important dynamical features. The currents in the study region are strongly impacted by a steep continental slope, which creates a strong ambient potential vorticity (PV) gradient. As a result, a meandering current—guided by the bathymetry—will cause a recurring FTLE pattern, with implications for both robustness and persistence. Dong et al. (2021) also identified persistent FSLE in the LoVe region. Although FSLE generally does not coincide with FTLE (Karrasch and Haller, 2013), we find agreement in our results with Dong et al. (2021), who further showed that FSLE features hinder cross-slope transport. In other words, the FTLE and FSLE fields both seem to have detected a dynamical transport barrier (if not perfectly impenetrable), which we believe should exist given the strong topographic PV gradient.

A caveat in analyzing FTLE fields in terms of their average is that details in shape and direction of individual FTLE ridges are lost. A ridge detection, that is the identification of what under ideal conditions will be LCS manifolds, could provide information on strain directionality, which may also be persistent under e.g. topographical current steering. Furthermore, the time averages smears out FTLE ridges to a large degree, even if their position only slightly changes in time. We expect FTLE ridges to be somewhat persistent along the continental slope, as the mean flow follows bottom topography, but due to small meandering over time these appear smooth and less persistent in $\overline{F}_t$. However, a meandering FTLE ridge is still affecting transport even though it moves slightly and does not keep its exact position over time. Inspecting the FTLE fields individually may reveal such meanderings, thus averaging over time may not be well suited for determining FTLE persistence.

Instead of averaging, one could follow a similar approach as Dong et al. (2021). Here, the authors defined a set of criterion for the existence of a particular FSLE ridge and investigated the frequency with which the criterion was fulfilled. Another approach could be to select a particular FTLE ridge and study how it evolves over time. Its lifetime, propagation distance, growth/dissipation rate, and structural evolution could then be studied. Possibly, a relationship could be established between the size, strength, and lifetime of the FTLE ridge.

Another method that has previously been used for investigating LCS persistence is the climatological LCS (cLCS) method described in Duran et al. (2018). Here, a velocity climatology is produced by averaging OGCM simulations over multiple years. This multi-year simulation may be considered a type of ensemble, where each one-year model simulation acts as an ensemble member. The velocity average removes flow fluctuations, thus highlighting persistent flow features, similarly to what we see in Figure 7. The cLCS are computed through a so-called quasi-steady LCS method, which may infer flow persistence and persistent transport barriers. However, we note that cLCS are not equivalent to hyperbolic LCS, and are thus not always describing a transport barrier. This method was later tested in the Brazilian current by Gouveia et al. (2020), where the authors state that large-scale flow features give rise to persistent quasi-steady LCS, coherent with our FTLE results (Figure 8). Even though the LCS-detection methods and settings are different, we see that the fundamental step to finding persistence through FTLE and cLCS is the ensemble average. Furthermore, in the presence of an EPS, a variation of the cLCS method may be adapted to handle EPS averaged velocities to study robustness of the fields. Likewise, a velocity climatology may be used to produce climatological FTLEs.

## 4.2 Temporal variability of FTLE

The analyses above confirm that the flow and the associated FTLE field are seen to vary drastically over short time periods. Flow features that develop pronounced structures in the FTLE field will drift, deform, and vanish over a range of time scales. Clearly, the lifetime of a particular FTLE ridge is restricted by the lifetime of the flow structure it represents. Specifically, features formed by large-scale circulation are more persistent, as these typically imply longer time scales.

Permanent geomorphological features present a defining constraint on the ocean circulation and, in particular, large-scale bathymetry steers ocean currents at high latitudes (Gille et al., 2004). In the LoVe region, the persistent topographically-steered NCC and NwAC give rise to frequent high-valued FTLE ridges along the continental slope, especially during winter. Individual FTLE ridges are hard to detect from the monthly-averaged FTLE field in Figure 4, but may be distinguished for shorter time averages where the smoothing effect due to averaging is smaller. The small-scale FTLE ridges, being more chaotic and short lived, are smoothed at the highest rate, whereas the large-scale FTLE anomalies remain visible for longer averaging times. On the other hand, computing the FTLE from an ensemble-averaged velocity field has proved to be more effective at isolating actual persistent FTLE ridges, as many small-scale flows which are highly variable over time are smoothed out from the velocity field.

Thus, time-averaging of FTLE fields can provide information about where FTLE ridges frequently form. However, we note that high values in the FTLE average may sometimes result from infrequent but very high FTLE values. Regardless, analysis of time-averaged FTLE is useful for identifying regions of material accumulation and entrapment. For instance, we expect that the semi-permanent anti-cyclonic Lofoten Vortex (Figure 1) in the middle of the Lofoten Basin (Raj et al., 2015; Isachsen, 2015) will form persistent and re-occurring FTLE ridges. Furthermore, submarine canyons in the LoVe region host a multitude of aquatic organisms, e.g. cold water coral reefs, which is possible because of the nutrient accumulation here (Sundby et al., 2013; Bøe et al., 2016). We speculate that these canyons will contribute to formation of persistent FTLE ridges, providing a control mechanism for particle transport towards specific locations.

### 4.3 Seasonal variability

Ocean currents in the LoVe region show seasonal variability in response to atmospheric forcing and the seasonally-varying hydrography. The autumn and winter months are characterized by westerly winds with transient low pressure systems passing through the region, and the water pile-up against the coast accelerates the currents. Spring and summer, in contrast, are dominated by moderate easterly winds (Furnes and Sundby, 1981) and weaker currents. In spring and summer, the seasonal stratification also responds to freshwater runoff and solar radiation (Christensen et al., 2018).

The associated seasonal ocean circulation patterns are reflected in the FTLE fields (Figure 5). Most pronounced is a clear difference in the intensity of the FTLE field over the continental slope. A well-mixed water column during winter results in more barotropic flow, hence the bathymetry controls the winter surface circulation. Thus, high FTLE values develop in the lateral shear region along topography-following slope currents. In contrast, seasonal stratification due to surface heating during summer leads to partial decoupling of the ocean surface layer from deeper currents. The result is that bathymetry has a weaker impact on surface flow structures during that season. Note that pronounced FTLE ridges may occur along the continental slope in summer, but these are less typical or weak, therefore tend to be washed out in both time and ensemble averages.

The coastline is expected to have a similar impact on FTLE formation throughout the year, as it directly affects surface currents during all seasons. However, around Moskstraumen (Figures 1 and 5), we identified higher FTLE variability in summer that is tied to tidal pumping through the narrow sound. It thus appears that surface-intensified flow, as a consequence of distinct summer stratification, may amplify surface currents and FTLE formation in this particular location (Sperrevik et al., 2017).

Finally, the fact that FTLE values are generally higher during winter can likely be interpreted in light of stronger atmospheric variability and forcing of the ocean during that season. In addition, more energetic flows at scales of 1–100 km can be generated by barotropic and baroclinic instability which, however, are also indirectly tied to stronger atmospheric forcing during winter (Callies et al., 2015).

### 4.4 Uncertainty of FTLEs in realistic flow fields

Ocean current uncertainty can be tied to non-linearities in the equations of motion: small errors in initial or boundary conditions, as well as tunable parameter values, can cause large errors in numerical integrations (Lorenz, 1963). Error propagation may thus have impacts in trajectory simulations (e.g. Zimmerman, 1986; de Aguiar et al., 2023). By this argument, uncertainties in FTLE fields derived from uncertain currents are expected. Here, we discuss the uncertainty of FTLE due to the flow field itself, but Allshouse et al. (2017) shows that windage has a clear impact on ocean surface LCS, which adds additional uncertainties.

A regional scale ocean EPS, Barents-2.5, is here used to describe uncertainties of ocean currents in the analysis and throughout the forecast range (Idžanović et al., 2023). By calculating FTLE fields for each ensemble member, we propagate the ocean model uncertainty into the FTLE analysis presented here. The various members exhibit differences in the FTLE fields, e.g. in terms of feature location, intensity and shape. Generally, FTLE ridges that exist in only one or few members are statistically unlikely to exist, emphasizing the need of an EPS when employing FTLE in operational oceanography.

Ensemble averaging is here suggested as a method to detect robust FTLE ridges, i.e. features that appear in a majority of members and can therefore be considered statistically likely to exist. Similarly to the time average, the ensemble average smooths out distinct features in the FTLE field, resulting in an average FTLE field $\overline{F}_m$ that highlights only the most predictable features. Some FTLE ridges can still be distinguished in $\overline{F}_m$, even after considering all 24 ensemble members of the Barents-2.5 EPS (Figure 6). In particular, high FTLE areas located along the continental slope tend to be more robust. As discussed above, the steep bathymetry plays an important role in causing the robustness, because even though the surface currents themselves are uncertain, the bathymetry constrains surface currents equally across the ensemble.

The spectral analysis (Figure 8) confirmed our expectations that large-scale FTLE ridges are more robust, as variability does not decay much at large scales when increasing the number of ensemble members in the averaging beyond 2–4 members. Small-scale features, however, are effectively removed by the ensemble average because they are more chaotic and exhibit lower predictability.

Sensitivity of the FTLE method has previously been investigated using satellite altimetry products by Harrison and Glatzmaier (2012), where the authors conclude that FTLEs are fairly insensitive to noise included in the velocity fields and that FTLEs are robust for large-scale eddies and strong jets. Gouveia et al. (2020) argues that persistent large-scale flows in particular give rise to quasi-steady LCSs, consistent with the persistent FSLE feature along the continental slope reported by Dong et al. (2021) which is also analyzed with FTLE in our study. In addition to time-persistent features, we investigate FTLE detection from transient flow features. Importantly, the Barents-2.5 EPS model used in this study can represent smaller and more transient structures than the aforementioned satellite products. From this we found that FTLE ridges are more uncertain at smaller scales, but robust where the flow is constrained by coastal or bathymetric steering.

## 4.5 Implications for operational forecasting

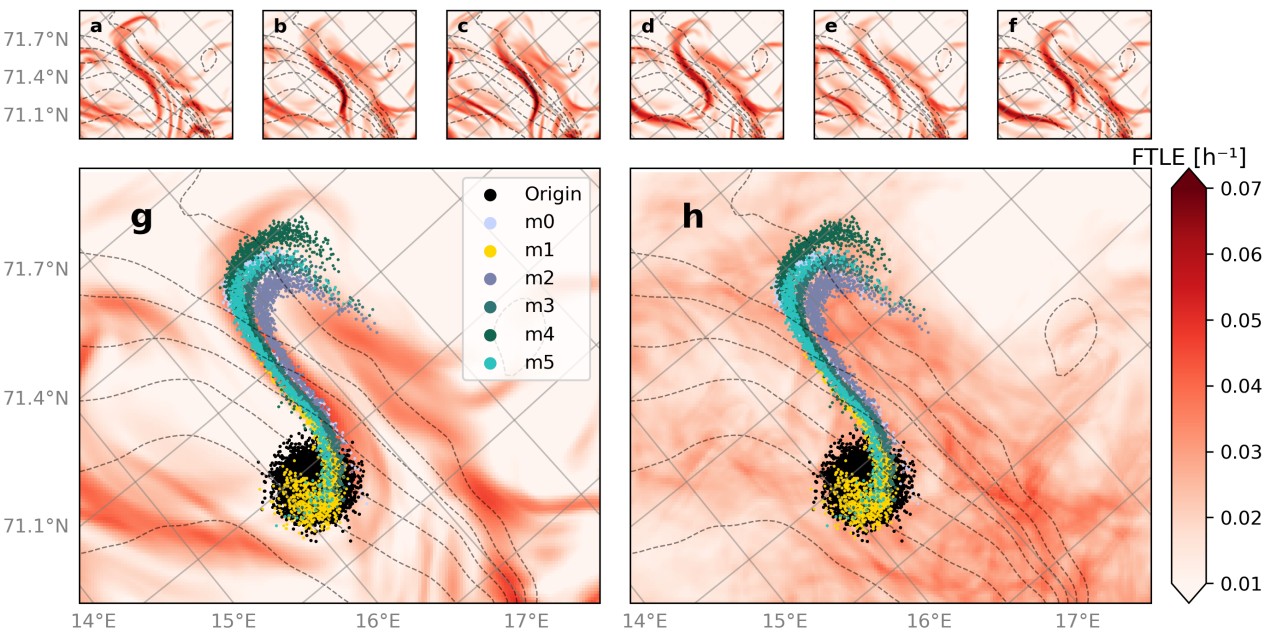

**Figure 9.** FTLE averages and particle clusters advected using velocity fields from a small number of different Barents-2.5 EPS ensemble members. Panels a)–f) show the 24-hour FTLE fields from each member used to advect the particles, while panels g) and h) show the ensemble-averaged FTLE field and the monthly averaged FTLE field from one member, respectively. Black dots in panels g and h mark the initial position of the particles on 2023-01-01, and their final positions after four days. Bathymetric contours are indicated with gray dashed lines

The main lesson from this study is that operational use of FTLE analysis in forecasting, e.g. for search-and-rescue, oil-spill operations or path-planning (e.g. Beegle-Krause et al., 2011; Ramos et al., 2018), should be viewed in context of the uncertainty in ocean current predictions and in light of the highly time-variable nature of the flow. Although FTLE fields are variable across an ensemble, we have seen that some features of the FTLE field are more robust than others. An ensemble of FTLEs must thus be assessed to separate robust from non-robust features. The detection of a robust FTLE ridge would thus present an opportunity to accurately identify search regions and dispatch environmental clean-up resources.

An approach to operationalizing a variation of the LCS analysis for search-and-rescue operations is described in Serra et al. (2020). Here, the authors discuss Objective Eulerian Coherent Structures (OECSs), originally introduced in Serra and Haller (2016), which identify attracting regions of the flow field and may be computed from a single snapshot of the velocity field, e.g. a satellite image or high-frequency radar measurement. The authors argue that the method is faster and provides a more complete coverage than Lagrangian particle trajectories, and show that this method is fairly robust to perturbations in the

underlying velocity field but are only valid for a short time. Although we use different methods, our findings are consistent with Serra et al. (2020) in that large-scale OECSs and FTLEs tend to be fairly robust to perturbations in the velocity field.

Much work on FTLE analysis and, more generally, on LCS detection lies ahead—also on the topics of robustness and persistence. But Figure 9 illustrates the underlying power of ensemble averaging. It shows a situation where there happens to be a high agreement between particle cluster trajectories over four days from a few different, randomly selected, ensemble members in the Barents-2.5 EPS. The ensemble-averaged FTLE field over all 24 ensemble members is also shown and appears to be highly robust: FTLE ridges remain clearly articulated in the average. More importantly, particle clusters from all ensemble members are seen to be attracted towards high values of $\overline{F}_m$. Thus, in such a case, the ensemble-averaged FTLE field provides a clear added value to trajectory forecasting in a real-time setting.

In contrast to this, the 30-day FTLE average from a single ensemble member, shown in Figure 9, does not shed much light on the short-term particle trajectories in this particular situation. This should not come as a surprise, as the FTLE ridges will have evolved substantially over the month. It is likely that certain FTLE ridges may be distinguished in shorter-term averages, taken over e.g. 3–4 days, could be utilized for short-term forecasting. However, in that case, it may be more appropriate to compute the FTLE field with $T = 3$ days instead, then obtaining the ensemble-averaged FTLE field over the time interval.

## 5 Conclusions

FTLEs are clearly imperfect representations of LCSs. And yet, FTLE analysis provides a practical diagnostic tool for analyzing how ocean-flow morphology associated with deformation impacts particle transport. In this numerical model study we have examined how the uncertainty of ocean model forecasts, illustrated in an ocean EPS, propagates into FTLE fields. It was shown that by employing ensemble averaging of FTLE fields, robust features of the FTLE field, that is features which the EPS system has gotten right in a statistical sense, may be separated from uncertain, non-robust, features. In particular, ensemble averaging typically retains flow structures at larger scales that are time evolving but predictable at specific times. The averaging will more typically wash out FTLE structures present in individual ensemble members, but it still has the potential to highlight regions over which FTLE ridges are statistically likely to emerge. Such features are often influenced by geomorphological constraints which, in our specific study region, was exemplified by a steep continental slope that imposes strong—and permanent—ambient PV gradients. We have also shown how such permanent environmental constraints can make FTLE fields persistent in time. So the over-all lesson learned from the study is that FTLE analysis can indeed add value to operational forecasting, even in light of the highly nonlinear and chaotic nature of real ocean flows. The key requirement is the forecast is treated as a probabilistic one, most practically produced using ensemble techniques.

*Code and data availability.* Archived data from the operational model runs of Barents-2.5 are disseminated on https://thredds.met.no/thredds/ fou-hi/barents_eps.html (Norwegian Meteorological Institute). Software for computing FTLE fields can be found on https://github.com/ mateuszmatu/LCS (Matuszak, 2024).

*Author contributions.* FTLE analysis: MM. Seasonal analysis: MM, PEI. Circulation model: MI, JR. Manuscript preparation: MM, JR, PEI, MI. Study concept: JR.

*Competing interests.* We declare absence of competing interests related to this work.

*Acknowledgements.* We thank the two reviewers, Rodrigo Duran and one anonymous reviewer, for very constructive criticism and great suggestions. We also acknowledge funding by the Research Council of Norway through grants 237906 (CIRFA), 300329 (EcoPulse), 314826 (TopArctic) and 314449 (Action).

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
