# Peer review of "Uncertainties in the Finite Time Lyapunov Exponent in an ocean ensemble prediction model"

_EGUsphere, 2024_

## Referee Comment (RC2)

This manuscript has inaccuracies, and shortcomings and lacks proper scientific context and some depth. However, the idea of ensemble-averaging and time-averaging to characterize LCS is a good one, and potentially, a good contribution. I will err on the side of the authors and recommend a major revision. My comments follow:

20 I would suggest you emphasize the sensitivity of trajectories and not the sensitivity of the velocity because even with a hypothetically perfect velocity, a small error in a trajectory's initial position can grow exponentially into large errors. The idea of your paper is to introduce perturbations in the velocity and measure how trajectories respond. Note the interest is in the trajectory response as visualized through LCS given a velocity ensemble spread. The main concern is trajectory uncertainty, even if explored in terms of velocity uncertainty.

In 35 you mention:

Previous studies often discuss the LCS methodology and their practical applications, but rarely touch upon the topic of LCS estimates being inherently affected by uncertainties in the velocity fields they aim to describe. Furthermore, short-lived flow features constantly develop, drift, and dissipate in real oceanic flow (Chen and Han, 2019). Given their time-dependency, LCSs might appear and disappear just as quickly. This brings up two important questions: (1) Given the velocity field uncertainty,

how robust, i.e. predictable, are LCSs derived from ocean models at a particular time?; (2) Given their time-dependency, how persistent are LCS in ephemeral flows?

I don't think robust can be equated with predictable. Robust in your study means that different realizations of a simulation (i.e. similar simulations) result in the same LCS. There is no predictive capability (i.e. estimates of future information based on past information) in this analysis

The short-lived structures you mention are not a problem, or even interesting, as it is straightforward to filter them and find the prominent deformation patterns, without the need to average see e.g. Olascoaga & Haller (2012; https://doi.org/10.1073/pnas.111857410) or Kunz et al (https://doi.org/10.5194/egusphere-2024-1215) that discusses the importance of persistence when it comes to attracting hyperbolic patterns and the lack of meaningful influence with short-lived structures (in particular while hyperbolic structures are forming or decaying). These are results without ensembles or any type of averaging. In particular, Olascoaga & Haller (2012) get rid of short-lived LCS by increasing the integration time T to 15 days. Thus, the choice of T= 1 day in your study raises the question of how do your results depend on your choice of T? Are the transient FTLE features that you filter through averaging unnecessarily increased by this choice? Wouldn't it be better to use a longer integration time to filter those features instead of time averaging? Also, as I will mention below, there are several papers showing 1) how to find persistent (or quasi-steady) LCS and 2) that persistent LCS are ubiquitous and meaningful. It is true however that we do want to be able to discern which are short-lived and not meaningful, and there is more than one way to get there.

Schematic 2 is not correct, the average of any number of zeros is still zero, i.e. regions where FTLE is zero in the left side of the schematic should also be zero on the right. Notice that in the caption of Figure 2 you introduce a concept that is not mentioned, or used, in any other part of the paper and that is "the average region covered by them". There must be a better way to convey the idea you want to convey.

40 should be Gulf Stream

40 The paper by Badza et al 2023 does not present reliable results (this is a paper that should have been rejected in my view) because:

They use a stochastic differential equation for the velocity to measure the robustness of LCS methods to noise. This is a big mistake. The mathematical theory of variational LCS explicitly states the results are only valid for a deterministic velocity, i.e. the results only hold for the typical ordinary differential equation dx/dt = v(x(t),t). This is a very basic, yet fundamental mistake that renders their results meaningless.

Even if the theory of hyperbolic LCS were to hold for a stochastic vector field, a stochastic component is not representative of the uncertainty typically encountered in a geophysical velocity field, neither simulated nor observed. An ensemble of simulations is a much better choice for uncertainty.

In their Gulf Stream case, they allow for a very long integration time (three months) while computing LCS within a limited spatial region, Thus the results are plagued by fictitious boundary effects, as is evident from their figures. A simple computation shows the inadequacy of their choice: Their domain is 30 degrees wide (note the flow is mainly west to east in their domain). That means their domain is less than 30*111=3330 km wide. Yet their integration time is 90 days, which means you would only need a velocity of 37 km/day (0.43 m/s) to traverse the whole domain, from west to east. The Gulf Stream commonly reaches a velocity of over 150 km/day (above 1.5 m/s).   Indeed, boundary effects in their results are apparent, and they mention it themselves: "Most of these streaks appear to look like diagonal lines, which is likely attributable again to the exodus of particles over the large period of flow considered."  They also mention in their discussion that: "As with most of the previously discussed methods, this can be attributed to the exodus of particles from the domain over our 90 day flow period." Even without the two mistakes mentioned above, there is not much that can be learned from results plagued by unphysical boundary effects.

In your study, Badwa et al. 2023 are cited to say that hyperbolic LCS detection is not reliable. However, as explained above their conclusion is meaningless. I therefore, as a reviewer, make the extraordinary suggestion that you delete, or at least adequately discuss, any sentence citing the Badza et al. paper, to avoid amplifying misleading results. The other papers you cite such as Harrison & Glatzmaier are better and are adequate for the point you wish to make regarding FTLE. In particular, the representation of uncertainty they choose is realistic and does not involve a fundamental dynamical-systems mistake.

55 what do you mean by dynamically active shelf region? Is there such a thing as a dynamically inactive sea? Either explain clearly the idea you want to convey or delete statements that don't add useful information, yet leave the reader wondering.

In the caption of Figure 1 you mention Moskstraumen has been indicated by an arrow, consider mentioning in the text what is this region. why is it important?

100 Although it is true that fluid parcels need to be advected, Equation 3 is not an accurate description of how Equation 2 is computed. It needs to be clear that you are time-integrating the velocity along a path which is not the same as integrating the velocity with respect to time at a fixed location, as your equation suggests. Importantly, you need to integrate two trajectories to be able to compute the distance \partial x, and it is not enough to just integrate the velocity as your equation reads.

105 Although deformation is indeed given by the singular values of the Jacobian of a Flow map, there is no such information as a "speed of deformation" embedded in the Flow map (note speed has units distance/time).

110 there is nothing to show, that is the definition of FTLE.

115 "Largest FTLE = LCS" is not true. This needs to be explained in detail throughout the paper so that your conclusions are not misleading. See my comments about strong FTLE produced by large horizontal shear in coastal regions in what follows.

125 "infinitesimally thin" is an unusual description. Although the width of a line within a plane indeed has measure zero, just like the width of a point within a line has measure zero, it is better to just say co-dimension 1 and leave it at that. In addition, co-dimension 1 is true for proper LCS, yet FTLE ridges tend to be coarse (as you suggest in your Figure 2 and other parts of the paper) and therefore your "infinitesimal" description is confusing. Best to omit this part.

Line 152, you mention larger FTLE at initial time is due to a large velocity gradient, this suggests high FTLE is produced by the velocity horizontal shear in which case it is not an LCS. Also, if it is an LCS then attraction rather than accumulation would be better.

Line 160 you claim longer averaging periods effectively decrease variability. Although this makes sense intuitively, it is hard for me to see this by just looking at the figures. For example, there does not seem to be a large difference between the 7-day standard deviation and the 28-day one. Can you quantify this further?

Lines 167-168 can you discuss further the relation between a persistent current and persistent FTLE? Why is not surprising that they co-locate? Is it due to velocity shear?

Figure 6, some colors are saturated (especially e and f) so we can't get a sense of how large the values are, also the mean and the std deviation are not too far off, consider plotting them with the same scale

(say 0.01 to 0.07 or whatever is needed so colors are not saturated over large regions). This will aid comparisons between mean and standard deviation.

172-173 It should be easy enough to test whether it is truly a transport barrier. How about releasing synthetic drifters on both sides of the barrier candidate and testing this directly? It would be nice to see results from individual members and some trajectory ensemble averages.

Figure 9a, the legends for daily winter and summer seem the same color. Also, why do the spectra for ensemble members (c and d) seem to decrease monotonically from member 1 at the top to the last member on the bottom?

Line 241 can you describe the dependence on the averaged members when only a few members are averaged?

Line 242, it would be nice to see the three regions used for the spectra, as you mention, some circulation patterns are highly predictable for example circulation along a slope tends to be quite predictable along large portions of the slope.

Line 243 and 252, could it be that FTLE seems to be more robust than persistent due to your choice of T=1day? Short T should be expected to result in more transient features relative to longer integration times. See for example the papers cited in the comments above for lines starting at 35.

Line 276, a repelling and attracting LCS cannot be parallel at the same location, if you go back to Dong et al you will see they describe an attracting LCS

277-278 and again we have the issue of T being relatively short at 1 day.

283 You mention other methods for detecting persistent LCS could yield more nuanced results. You also cite (line 343) a paper by Gouveia et al to say that large-scale features give rise to quasi-steady LCS. If you read the Gouveia et al paper carefully you will see they use a method to find quasi-steady LCS that was published in 2018 (https://doi.org/10.1038/s41598-018-23121-y). This later paper has over 40 citations according to Google Scholar, suggesting that many other studies are using that method to extract quasi-steady LCS. This topic is directly relevant to your study, so it seems your literature review is lacking. As you will see throughout my comments, the use of FTLE is a problematic issue that keeps coming up. The methodology published in 2018, and used by Gouveia et al, does not rely on FTLE, although there is some averaging. The difference in approach suggests a worthwhile discussion regarding the differences between that approach and your approach, including the strengths and weaknesses of each method.

288 it is also possible that a very strong FTLE shows up in the average, even if it does not persist much in time, especially if it recurs.

358-359 Strong, persistent FTLE can be caused by persistent horizontal shear in which case it would not be indicative of LCS. Strong persistent FTLE can be expected in many coastal regions to be caused by horizontal shear. High FTLE next to the coastline, as in Fig. 5 during the summer or around 69.4N in the winter, for example, should be particularly suspect.

You need to clarify throughout your paper that strong FTLE may be caused by horizontal shear, in which case FTLE is NOT indicative of an LCS, and mention that horizontal shear can be persistently high at certain locations such as a coastline. These locations, according to your suggestion, would have persistent LCS due to the persistent high FTLE, yet FTLE is not indicative of LCS if it is solely due to shear.

It is not enough to mention FTLE ridges only approximate LCS and to reference where to find the distinction between the two (line 144), because this distinction directly impacts the interpretation of our results, as has been explained above.

382 "…by combining LCS analysis with ensemble prediction methods." The 2018 method to find quasi-steady LCS mentioned above should be discussed in this context as well. Can that method be used to find robust features in operational forecasting? Or is it complementary information to the ensemble methods you propose? Or are these two methods for differing purposes? Can you expand on how to use these methods to detect robust or persistent LCS in terms of operational oceanography? How concretely can these methods be applied? Can you suggest step-by-step instructions on how to implement these methods in an operational application? Can you give an example of how they have been used or can be used in operational oceanography? How feasible, useful, and accessible are these methods in operational oceanography?

---

## Author Comment (AC1)

**Answer to reviewer #1**

Thank you for your detailed comments, which highlight a number of fundamental inaccuracies as well as more subtle issues which we are prepared to address in a revised manuscript. Initial responses to your various points are given below:

*1. The term "Lagrangian Coherent Structures" (henceforth LCSs) is used prominently in the title, and throughout the article. This term was coined by Haller in the early 2000s, and it is the continuing work of this group that the authors almost exclusively cite in this paper. However, the work of this group now very clearly defines LCSs in very specific ways, which are outlined clearly in Haller (Annu Rev Fluid Mech, 2015). These definitions, stated within the two-dimensional context in which the current paper is situated, relate to attempting to find curves towards/from there is extremal attraction/repulsion -- so-called "hyperbolic LCSs." It is these, and some allied entities, that are LCSs according to Haller and the group's definition, and so citing those papers and not using those definitions does not make sense. Moreover, Haller and collaborators in this and a range of other papers make clear that FTLEs are not necessarily these LCSs, and so prominently using the term LCSs in this paper is incongruous at best. Since the paper deals exclusively with FTLEs (and their averaging -- not even their ridges which some other authors may refer to as a type of LCS), the title and the paper should exclusively use the term FTLEs. Of course, one should position FTLEs in the context of LCSs -- which to a range of other authors represent an entire suite of techniques devised to extract coherent structures (based on differing definitions and intuition) from genuinely unsteady data. There are many recent reviews of this range of different methods for LCSs in this broader context in the literature: Balasuriya et al (Physica D, 2018), Hadjighasem et al (Chaos, 2017), Shadden (in: Transport and Mixing in Laminar Flows, Wiley, 2011, pp59). The authors are advised to consult these in positioning their work (as in line 85 when they say "Various methods have been proposed for LCS detection") and deciding whether the term LCS is actually appropriate.*

We agree that in this manuscript, we are dealing with the variability of FTLE fields which are not directly equivalent to Lagrangian Coherent Structures. Furthermore, we realize that there are some fundamental issues with the FTLE as a tool for LCS detection, and that more modern methods exist. We intend to clarify that in the manuscript, discuss the weaknesses of FTLE with regards to LCS, compare it to other existing methods and adapt the manuscript's title accordingly.

*2. The main conclusion seems to be that taking FLTEs and averaging them gives a better diagnostic of "robust" coherent structures. In doing such an averaging, there are some scientific issues related to time-parametrization and that I will come back to in a later point. However, I have some comments with respect to the issue of averaging FTLE fields to smooth out what the authors seem to think of as non-robustness, and thereby extracting robust structures. First, the FTLE fields are definitely associated with a time-of-flow, and since the authors use 24 hours exclusively here, they will be identifying local exponential stretching rates over the past 24 hours. If the authors were \*not\* interested in stretching over that time-scale but rather over, say, a one-month period, then rather than taking a 24 time-window, they should take a one-month window. This will definitely smear over ephemeral stretching. (By the way, when the authors use the term "ephemeral" this depends*

*on the context. It seems that this means structures which do not persist over a longer time-scale, say a week or two. Within this time-frame, perhaps 24 hours is ephemeral -- but the authors seem to have decided to use 24 hours in the FTLE calculation by choice, and so it's to be expected that features related to 24-hour time scales are what will be revealed in the FTLE field.) Second, the discussion seems to indicate that the authors want to find stationary structures which persist over a longer time. In other words, to think of the velocity field as predominantly steady, with unsteady variations, and the goal would be to find structures which are stable with respect to the unsteadiness (which is assumed smaller). Well, if so, there are better ways to approach this. Rather than calculating FTLE fields from the full unsteady Eulerian data, they can obtain a dominant steady part of the Eulerian velocity. One way to do this would be to average the Eulerian data over an explicit time period -- this has the added advantage of being able to specify a time-scale (the time of averaging) over which the Eulerian field is assumed "mostly" steady, and hence one can ascribe a time-scale to one's conclusions. Another alternative would be to use some smoothening technique -- and again, if using something like a spatial filter associated with an explicit length-scale, one can ascribe a scale (a length-scale in this instance) of "accuracy" of the processed data to enable any conclusions to be stated in relation to that. These methods work directly on the Eulerian velocity field, rather than applying techniques such at FLTEs on data which to all intents and purposes (based on the conclusions reached) seem to be viewed as "noisy." Remove the noise first to avoid amplification of inaccuracies when doing additional computations. Thirdly, if stationary objects are sought, it seems that the authors want to look at the dominant steady component, and if one has a steady velocity field, Lagrangian methods are irrelevant. Lagrangian issues only make sense if there is unsteadiness, and one needs to follow the flow. If steady, simple techniques (drawing streamlines, Okubo-Weiss criterion, etc) on the Eulerian velocity field give perfectly good information on what's going on.*

We realize that the choice of time (the integration time) is an issue, and we addressed this in comment #4. However, our intention was not to find structures which exist over longer time periods (e.g. longer than 24 hrs). In our averaging in time over consecutive FTLE fields we sought to identify regions where high values in the FTLE appear *frequently*, fully aware of the fact that individual structures will form, drift, and deform over various time scales. We will clarify this in a revised manuscript. We appreciate your suggestions on how we can approach this issue, and intend to investigate the proposed simple techniques to see how these compare to FTLE.

*3. Based on the conclusions that the authors seem to be reaching, there seems to be an incomplete understanding of what the FTLEs represent. The FTLE field $ \sigma_{t_0}^t $ represents a field at time $ t_0 $, associated with the stretching rate experienced by infinitesimal fluid parcels beginning at time $ t_0 $ and flowing until time $ t $. This rate is converted to an exponential one via the logarithm, and the time-of-flow $ T = |t-t_0| $ is used to time-average this quantity so that it's the average exponential rate over the time period considered. Note that there is absolutely no mention here of anything being a flow barrier, or a "coherent structure." Thus, the FLTE identifies regions in the time $ t_0 $ based on stretching rates over the time-of-flow (in this case, 24 hours in the past). When one time-averages the FTLE field, what exactly does that mean? Presumably (and this is not clearly stated), FTLE fields are generated for differing $ t_0 $s but the same time of flow (this is sometimes called time-windowing in the coherent structure community), but then are the*

*authors averaging over $t_0$? (Give an explicit formula for the averaging, so that this is clear.) If so, they are taking scalar fields which are defined over different times $t_0$, each of which is associated with flow over a different time-window $(t_0-T,t_0)$, and averaging them. If this is what is done, it needs to be explained clearly. But then, the interpretation of this needs to be carefully stated, since one is averaging over different initial times, and what one gets cannot be associated with a particular time instance (unlike one calculation of an FLTE field, which gives a field at time $t_0$). Of course, one might argue that calculationally the averaging tells us of how 24-hour motion calculated over several initial times (say 1 January to 31 January 2023) is averaged to give an "average exponential rate of motion in January 2023 when a time-scale of 24 hours is considered for the exponential rate". This would need to be explained, because it's quite awkward and hard to interpret.*

In a revised manuscript we will tone down the interpretation of FTLE maxima as transport barriers, especially in the core of the paper. We will instead expand the introductory section with more mention of previous works that investigate this relationship—and then return to the issue in the Discussion/Conclusions section. We will also be more careful about how we define FTLE and how it describes the flow field. Finally, we also plan to elaborate on the methods used to conduct the time averaging, so that it becomes clear to the reader. Several of the reviewer's ways of formulating these definitions and interpretations are actually right to the point and will form the basis for revised text.

*4. The above point related to one aspect of time-parametrization: the $t_0$ in the field $\sigma_{t_0}^t$. Another time-parametrization issue is the $t$, and thus the time-of-flow. Everything in this paper has used a time-of-flow of 24 hours. Hence, everything is slaved to this time-scale -- the exponential rate is computed based on time-of-flow for this time-scale. This issue is buried in the paper, with the multitude of plots not mentioning this explicitly If a time-of-flow of 48 hours were chosen instead, how do things change? Basically, the calculation of an FTLE field explicitly picks out a time-scale, and this is not something which the authors have clarified. The results are explicitly associated with this time-scale, and no other. If the results are to be used in forecasting, why is this the correct time-scale? Or is this method robust to changing the time-scale? How does the time-scale interact with the time associated with the time-averaging as discussed in my previous point? Basically, the issue of TIME (initial plus time-of-flow in calculating the FTLE and the appropriate interpretation of the FTLE field, the times of computation chosen for averaging) is crucial, and needs to be carefully explained, interpreted, and robustness evaluated (if appropriate).*

The choice of 24 hours as integration period is not completely arbitrary but motivated by typical uses of ocean forecasting models. These are decision support tools for search-and-rescue operations, oil-spill modeling, ice-berg trajectory forecasts, and similar trajectory analysis which often require forecasts of a few hours up to a few days. But the comment is definitely warranted, and we will now provide an analysis figure with examples of various integration lengths for the FTLE (time-of-flow). The figure (in its current form) is added here, and it indicates that our results are not overly sensitive to the integration period within integration length of 12 hours to 72 hours. But we do see (and will discuss) some interesting differences. For example, for much longer integration periods of 15 to 30 days, we see that the FTLE analysis yields distinct linear features in low current velocity regions, especially in the deep basins off the continental slope In contrast, in the energetic flow regions over the shelf and slope, longer integrations tend to smear out features. Our

interpretation is that the ability of the FTLE field to pick up LCSs (in a broad sense) depends on the integration period matching the time scale of the dynamics—which varies depending on the environmental conditions.

We also compare FTLE analysis based on long integration periods with the time-average of several short integration periods and see distinct differences in the two approaches. Wheres the time-average provides a description of regions that are typically abundant with FTLE ridges, the long-time integrations do not distinguish areas in the high-velocity region but instead allow to better characterize low-velocity regions, which are less pertinent to time-critical contingency modeling.

We do not take an opinion on right or wrong time integration, but in the revised manuscript we like to discuss its impact on the analysis, and provide a view on how an appropriate time period may be selected depending on the application in focus. In our initial analysis, we did indeed experiment with different periods from one hour up to a few days, and some of these examples are presented below (see Fig. A).

*5.  Returning to the definition of the FTLE mentioned earlier.  It represents a field at time $t_0$, associated with the average exponential rate over flow from time $t_0$ to $t$.  Note that there is absolutely no mention here of anything being a flow barrier, or a "coherent structure."  Yes, there are papers in the literature which seem to indicate such a connection, but the reality is that it is unjustified.  The early papers in this area used STEADY toy models which had saddle points with one-dimensional stable and unstable manifolds emanating from them, and since these manifolds are associated with exponential decay rates, came up with the idea that FTLE ridges had something to do with stable and unstable manifolds.  And these manifolds were flow barriers in some way.  However, this argument does not hold water, since there are examples such as in Haller (Physica D, 2011) which show that the stable/unstable manifold interpretation sometimes fails even in infinite-time flow.  (And, getting back to a previous point related to hyperbolic LCSs, the fact that repelling/attracting do not necessarily occur as expected are also shown.)  Real data is much worse: it is unsteady, and finite-time.  Finite-time aspects are awkward for inferring exponential rates of growth, since any function over a finite-time can be bounded by an exponential. Unsteadiness is yet another problem, because (even in the infinite-time context) saddle points generalize to hyperbolic trajectories, and their stable/unstable manifolds move around (another reason why time-averaging is questionable).  Furthermore, it is not clear what an "FTLE ridge" is -- one never gets a genuinely one-dimensional curve which is well-defined, but rather gets regions of larger FTLE values.  Balasuriya et al (J Fluid Mech, 2016) provide an assessment and interpretation of what the FTLE means, with an emphasis on fluid motion, which helps understand these issues.  In particular, for finite-time, unsteady data sets, using FTLEs and their ridges cannot easily reach conclusions regarding flow barriers and coherence.  FTLEs explicitly look at exponential growth rates with respect to the time-of-flow considered, and that's about it.  So when one takes FTLE fields, as done in this paper, and tries to reach conclusions regarding coherent structures such as eddies (as the authors do towards the end of the paper), this needs to be treated with suspicion, because it is not on any firm scientific grounds.  The idea that eddies can be demarcated by FTLE ridges -- notwithstanding my earlier comments on finite-time and unsteadiness -- may go back to work in the 1990s (such as del-Castillo-Negrete, Knobloch, Pierrehumbert) who analyzed perturbed toy models.  In these cases, the unperturbed models were explicit and steady, and had saddle points with stable and unstable manifolds.  In some cases, the geometry was such that these manifolds encircled an eddy.  Since the manifolds many be*

*discoverable using FTLE ridges (again subject to various caveats), in such cases ONLY, one might think of an eddy as being found using FTLE fields. However the interior of the eddy does NOT typically contain exponential stretching, and hence FTLE fields by themselves cannot be used to reliably identify eddies as the authors here are doing in their later figures. Actually, in the standard fluid-mechanical dichotomy between stretching and rotation, the eddies have the opposite of stretching, and thus the FTLE is exactly the wrong thing to use (Okubo-Weiss and related criteria may help; however as noted by many authors unsteadiness is a problem in using such Eulerian characteristics). Basically, statements such as "two additional regions are identified and considered as robust in Fig.7 " (line 211 in the paper) are, in this vein, questionable. Indeed, high FTLE values indicate greater separation, and hence LESS certainty!*

As outlined in relation to an earlier reviewer comment, we intend to rework the manuscript to clarify that FTLEs and LCSs are not the same thing, and focus on the properties of the FTLE only. We thank the reviewer for providing the paper by Balasuriya et al (J Fluid Mech, 2016), which we intend to use when discussing FTLE. We use the term "FTLE ridges" loosely to discuss "curves" formed by locally high FTLE values which can be seen in our figures of the FTLE field. However, we agree that we don't get a genuine and well-defined 1D curve in the FTLE field, and will make changes in the manuscript to reflect this.

We also agree that it is unjustified to identify eddies through the FTLE, which was never intended as the main focus in this manuscript. The original purpose of identifying the eddies was that these are easily identifiable flow structures appearing in the velocity data, such that they could be used to infer something about FTLE ensemble variability. We see that this point is not clear, and instead it seems that we are attempting to use the FTLE to identify eddies. In a revised manuscript, we will focus less on the eddies and clarify any misconceptions written about FTLE regarding eddies.

*6. The impression given throughout is that the authors are examining robustness of LCSs. I've already talked about why it's actually FLTEs and not LCSs, but in this point I want to question whether robustness is the right thing. There are many recent papers which examine robustness of FTLE fields (Balasuriya, J Comp Dyn, 2020; Guo et al, IEEE Trans Visual Comp Graphics, 2016; Raben, Exp Fluids, 2014), but this paper is not one of them. (There are a few other papers, from oceanographic situations, which the authors speak to in lines 339-344.) By "robustness," the authors seem to mean smearing over small time scale motion, in other words looking for entities which persist over longer times. This needs to be made clear throughout the manuscript. (I've talked about the time-parametrization issue previously; to smear over smaller time scales, one needs to simply choose appropriate times for the FTLE which are relevant to what one is looking for.) In Section 4.3, also, the word "uncertainty" is questionable because of this same reason -- the authors have no calculated any uncertainty (i.e., have not evaluated anything to do with uncertainty in the input data).*

We would like to take this opportunity to clarify that we suggest a distinction between 'robustness' and 'persistence' of FTLE fields as calculated from ocean model forecasts. Much of the discussion in literature regards what we identify as persistence, i.e. time variability. In our discussion, robustness refers to similarities of FTLE features across an ensemble of model flow realizations. We are aware of the studies which address uncertainties in FTLE computations, and will discuss them in our revised manuscript. Our goal was to investigate this issue from an operational perspective and investigate how an

FTLE field manifests itself in an ensemble prediction system, directly addressing uncertainty in forecasts.

We plan to clarify this distinction and put less emphasis on time variability as this subject has been more extensively been addressed in other studies. Furthermore, the reviewer suggested including an analysis of current velocities, which we would like to adopt in a revised manuscript to show how variability in the flow field translates into variability in FTLE fields.

*7.  There are several statements around Equation (2) and (3) which are incorrect. The statements in line 101 are all incorrect: $ \delta x $ is not the final location, but $ x $ is, and the (1,1) term in Equation (2) then represents the partial derivatives of $ x $ with respect to $ x_0 $, for example.  The integral bounds in (3) are unclear and inconsistent with the previous equation.  Presumably the authors mean something like $$ x(t) = x_0 + \int_{t_0}^t u \left( x(\tau), t(\tau) \right) \mathrm{d} \tau $$, and then one needs to also clarify that $ x(\tau),y(\tau) $ are the evolving trajectory locations.  However, this integral formulation may not be the most natural.  The differential form with $ \dot{x} = u \left( x(t), y(t) \right) $, $ \dot{y} = v \left( x(t), y(t) \right) $ and the initial condition $ \left( x(t_0), y(t_0) \right) = \left( x_0, y_0 \right) $ connects better with the discussion, perhaps.*

We will look through our equations in section 2.3 again and fix mistakes.

*8. The discussion at the end of Section 2.3 is fraught.  What is a "2D curve" [line 126]?  The statement that "averaging smooths the ridges into fields of attraction/repulsion" [line 128] is incorrect because, as mentioned previously, claiming that FTLEs have anything to do with attraction/repulsion is questionable.  "Making these ridges more certain" [line 130] relates to an earlier comment that what is being done here has nothing to do with robustness or certainty.  The authors comment that the time- and ensemble-averaged FTLE fields are not transport barriers (which is correct -- but neither is the FTLE field -- and again it's not clear to me how a field can be a barrier), but then talk about things being barriers over larger regions.  Figure 2 is strange.  One does not usually get FTLE ridges (even in idealized steady toy models in 2D) which intersect and pile on each other like this.  Intersections in such cases occur at saddle points.  If unsteady, intersections that one gets, analogous to intersections between stable and unstable manifolds, can relate to chaotic motion -- but these are between forward-time FTLE ridges and backwards-time FTLE ridges, rather than self-intersections within one of these.*

We acknowledge that we have been too quick on equating high values in the FTLE field to attraction or repulsion, and that we used the term "transport barriers" too loosely. We have in fact only looked at the FTLE field, and not conducted any ridge detection or investigated further criteria from e.g. Farazmand and Haller (2012) to distinguish any potential LCS. Furthermore, as the second reviewer pointed out, strong values in the FTLE field may be produced by horizontal velocity shear, which does not result in attraction/repulsion. We intend to conduct a major revision of the manuscript, and downplay or question  the term "transport barriers". We do, however, believe there is such a thing as a dynamical transport barrier, specifically related to the strong potential vorticity (PV) gradient associated with a steep continental slope (as we have in our domain). A PV barrier is not impenetrable, but it does inhibit tracers and particles - as documented throughout the dynamical literature (we will add references). And we do believe that the high FTLE values over the continental slope,

also when averaging over time or over the ensemble, do in fact arise from the strong PV gradient (or 'PV barrier' if one allows a more loose use of the term). So, in the revised manuscript, we will be more careful in explaining how we use the term, aware that different parts of the research community might be using various degrees of rigor in their definitions.

We agree that Fig. 2 is misleading and that structures like these do not appear in nature. The idea behind Fig. 2 was to show what FTLEs might look like over the ensemble dimension, where each drawn line represents FTLE ridges from different ensemble members. This is not shown clearly enough in neither the figure nor the text, and will be reworked.

*9. The fact that a large standard derivation of the time-averaged FTLE usually is close to where the FTLE is large [line 169-171] is no surprise. Large FTLE relates to larger uncertainty in the results, because any errors (based on interpolation to a grid, say) increase exponentially. Hence, the values one assigns to the FTLE tend to be less certain. This issue is well-known, and described in some of the papers I've mentioned previously on robustness of FTLEs.*

Thank you for this remark. This will be considered when reworking the manuscript.

*10. For the discussion on Section 3.4, I can't quite understand what the $k$ in the figures is. Since in 2D, one has two wavenumbers -- say $l$ and $m$, associated with the eastward and northwards coordinates. I don't understand the discussion [lines 230 onwards] about averaging over rows and columns (over $l$ and $m$?). Is $k = \sqrt{l^2 + m^2}$? The spectral plots in Figure 9 are used to infer robustness in some way, based on the fact that one gets decay with $k$ in the bottom figures, say. Any smoothening process will of course get rid of the smaller wavenumbers. Similarly one expects more smoothness when averaged over more and more days.*

The procedure for producing Fig. 9 has been poorly described in the manuscript. The transformation of the FTLE field to Fourier space has been done using the 2D discrete cosine transformation method. As pointed out, this results in two wavenumbers, m and l. These are combined to create a radial wavenumber k = sqrt(m^2 + l^2). We intend to revise this section, include equations and references, and clarify it in the discussion.

[Figure]

Fig. A: Backwards FTLE for different integration times (6, 12, 24, 48, 72, 168, and 672 hours). The final panel shows the monthly average of 24-hours long FTLE computations.

---

## Author Comment (AC2)

**Answer to reviewer #2**

We greatly appreciate your detailed reading and your comments on our manuscript. You have requested a number of clarifications, corrections and to some extent further analysis and discussion, all of which we are prepared to undertake.

*20 I would suggest you emphasize the sensitivity of trajectories and not the sensitivity of the velocity because even with a hypothetically perfect velocity, a small error in a trajectory's initial position can grow exponentially into large errors. The idea of your paper is to introduce perturbations in the velocity and measure how trajectories respond. Note the interest is in the trajectory response as visualized through LCS given a velocity ensemble spread. The main concern is trajectory uncertainty, even if explored in terms of velocity uncertainty.*

This is a good point and we will adapt this change in wording.

*In 35 you mention:*

*Previous studies often discuss the LCS methodology and their practical applications, but rarely touch upon the topic of LCS estimates being inherently affected by uncertainties in the velocity fields they aim to describe. Furthermore, short-lived flow features constantly develop, drift, and dissipate in real oceanic flow (Chen and Han, 2019). Given their time-dependency, LCSs might appear and disappear just as quickly. This brings up two important questions: (1) Given the velocity field uncertainty, how robust, i.e. predictable, are LCSs derived from ocean models at a particular time?; (2) Given their time-dependency, how persistent are LCS in ephemeral flows?*

*I don't think robust can be equated with predictable. Robust in your study means that different realizations of a simulation (i.e. similar simulations) result in the same LCS. There is no predictive capability (i.e. estimates of future information based on past information) in this analysis*

We agree that predictability does not directly follow from robustness. This would only be the case if the underlying circulation model is proven to have both predictive skill on both the time and the spatial scales of interest, and if the ensemble spread exhibits reliability. Both criteria are valid only to a limited extent in terms of surface currents, which has been investigated by Idzanovic et al. (2023) for the model system that is used here. Our study is interested in whether the EPS yields similar FTLE fields at a specific time, and does not focus on their predictability. This point will be clarified in the manuscript.

*The short-lived structures you mention are not a problem, or even interesting, as it is straightforward to filter them and find the prominent deformation patterns, without the need to average see e.g. Olascoaga & Haller (2012; https://doi.org/10.1073/pnas.111857410) or Kunz et al (https://doi.org/10.5194/egusphere-2024-1215) that discusses the importance of persistence when it comes to attracting hyperbolic patterns and the lack of meaningful influence with short-lived structures (in particular while hyperbolic structures are forming or decaying). These are results without ensembles or any type of averaging. In particular, Olascoaga & Haller (2012) get rid of short-lived LCS by increasing the integration time T to 15 days. Thus, the choice of T= 1 day in your study raises the question of how do your*

*results depend on your choice of T? Are the transient FTLE features that you filter through averaging unnecessarily increased by this choice? Wouldn't it be better to use a longer integration time to filter those features instead of time averaging? Also, as I will mention below, there are several papers showing 1) how to find persistent (or quasi-steady) LCS and 2) that persistent LCS are ubiquitous and meaningful. It is true however that we do want to be able to discern which are short-lived and not meaningful, and there is more than one way to get there.*

Reviewer #1 raised a similar concern about the integration period. The choice of 24 hours as integration period is not completely arbitrary but motivated by typical uses of ocean forecasting models. These are decision support tools for search-and-rescue operations, oil-spill modeling, ice-berg trajectory forecasts, and similar trajectory analysis which often require forecasts of a few hours up to a few days. But the comment is definitely warranted, and we will now provide an analysis figure with examples of various integration lengths for the FTLE (time-of-flow). The figure (in its current form) is added here, and it indicates that our results are not overly sensitive to the integration period within integration length of 12 hours to 72 hours. But we do see (and will discuss) some interesting differences. For example, for much longer integration periods of 15 to 30 days, we see that the FTLE analysis yields distinct linear features in low current velocity regions, especially in the deep basins off the continental slope In contrast, in the energetic flow regions over the shelf and slope, longer integrations tend to smear out features. Our interpretation is that the ability of the FTLE field to pick up LCSs (in a broad sense) depends on the integration period matching the time scale of the dynamics—which varies depending on the environmental conditions.

We also compare FTLE analysis based on long integration periods with the time-average of several short integration periods and see distinct differences in the two approaches. Wheres the time-average provides a description of regions that are typically abundant with FTLE ridges, the long-time integrations do not distinguish areas in the high-velocity region but instead allow to better characterize low-velocity regions, which are less pertinent to time-critical contingency modeling.

We do not take an opinion on right or wrong time integration, but in the revised manuscript we like to discuss its impact on the analysis, and provide a view on how an appropriate time period may be selected depending on the application in focus. In our initial analysis, we did indeed experiment with different periods from one hour up to a few days, and some of these examples are presented below (see Fig. A).

*Schematic 2 is not correct, the average of any number of zeros is still zero, i.e. regions where FTLE is zero in the left side of the schematic should also be zero on the right. Notice that in the caption of Figure 2 you introduce a concept that is not mentioned, or used, in any other part of the paper and that is "the average region covered by them". There must be a better way to convey the idea you want to convey.*

We agree that Fig. 2 is misleading and that structures like these do not appear in nature. This is also a concern that was raised by reviewer #1. The idea behind Fig. 2 was to show what FTLEs might look like over the ensemble dimension, where each drawn line represents FTLE ridges from different ensemble members. This is not shown clearly enough in neither the figure nor the text, and will be reworked.

*40 should be Gulf Stream*

This will be fixed.

*40 The paper by Badza et al 2023 does not present reliable results (this is a paper that should have been rejected in my view) because:*

*They use a stochastic differential equation for the velocity to measure the robustness of LCS methods to noise. This is a big mistake. The mathematical theory of variational LCS explicitly states the results are only valid for a deterministic velocity, i.e. the results only hold for the typical ordinary differential equation dx/dt = v(x(t),t). This is a very basic, yet fundamental mistake that renders their results meaningless.*

*Even if the theory of hyperbolic LCS were to hold for a stochastic vector field, a stochastic component is not representative of the uncertainty typically encountered in a geophysical velocity field, neither simulated nor observed. An ensemble of simulations is a much better choice for uncertainty.*

*In their Gulf Stream case, they allow for a very long integration time (three months) while computing LCS within a limited spatial region, Thus the results are plagued by fictitious boundary effects, as is evident from their figures. A simple computation shows the inadequacy of their choice: Their domain is 30 degrees wide (note the flow is mainly west to east in their domain). That means their domain is less than 30\*111=3330 km wide. Yet their integration time is 90 days, which means you would only need a velocity of 37 km/day (0.43 m/s) to traverse the whole domain, from west to east. The Gulf Stream commonly reaches a velocity of over 150 km/day (above 1.5 m/s). Indeed, boundary effects in their results are apparent, and they mention it themselves: "Most of these streaks appear to look like diagonal lines, which is likely attributable again to the exodus of particles over the large period of flow considered." They also mention in their discussion that: "As with most of the previously discussed methods, this can be attributed to the exodus of particles from the domain over our 90 day flow period." Even without the two mistakes mentioned above, there is not much that can be learned from results plagued by unphysical boundary effects.*

*In your study, Badwa et al. 2023 are cited to say that hyperbolic LCS detection is not reliable. However, as explained above their conclusion is meaningless. I therefore, as a reviewer, make the extraordinary suggestion that you delete, or at least adequately discuss, any sentence citing the Badza et al. paper, to avoid amplifying misleading results. The other papers you cite such as Harrison & Glatzmaier are better and are adequate for the point you wish to make regarding FTLE. In particular, the representation of uncertainty they choose is realistic and does not involve a fundamental dynamical-systems mistake.*

Thank you for this comment. We did not realize this issue with Badza et al. (2023), and intend to remove the citation from our manuscript.

*55 what do you mean by dynamically active shelf region? Is there such a thing as a dynamically inactive sea? Either explain clearly the idea you want to convey or delete statements that don't add useful information, yet leave the reader wondering.*

What we meant here is that the shelf and slope region has more variability due to the presence of a complex coastline and the juxtaposition of different water masses (setting up frontal zones) that generate various forms of flow instability. We will clarify in a revised manuscript.

*In the caption of Figure 1 you mention Moskstraumen has been indicated by an arrow, consider mentioning in the text what is this region. why is it important?*

The velocity field through Moskstraumen is highly tidally dependent, and in a previous iteration of our manuscript, we investigated whether this yields periodic and thus predictable FTLE fields. As this is not included in the submitted manuscript we intend to update Fig. 1 and remove the arrows indicating Moskstraumen.

*100 Although it is true that fluid parcels need to be advected, Equation 3 is not an accurate description of how Equation 2 is computed. It needs to be clear that you are time-integrating the velocity along a path which is not the same as integrating the velocity with respect to time at a fixed location, as your equation suggests. Importantly, you need to integrate two trajectories to be able to compute the distance \partial x, and it is not enough to just integrate the velocity as your equation reads.*

We will take a look at Eq. 3 again and explain better how to obtain \Delta F.

*105 Although deformation is indeed given by the singular values of the Jacobian of a Flow map, there is no such information as a "speed of deformation" embedded in the Flow map (note speed has units distance/time).*

This will be fixed.

*110 there is nothing to show, that is the definition of FTLE.*

This will be fixed.

*115 "Largest FTLE = LCS" is not true. This needs to be explained in detail throughout the paper so that your conclusions are not misleading. See my comments about strong FTLE produced by large horizontal shear in coastal regions in what follows.*

The other reviewer also commented on this. We realize that the largest FTLE does not necessarily imply an LCS and that there are additional criteria which an FTLE ridge must satisfy to qualify as an LCS approximation. We intend to clarify this difference between FTLE and LCS in a revised manuscript, including remarks about the criteria needed to detect LCS from FTLE fields.

*125 "infinitesimally thin" is an unusual description. Although the width of a line within a plane indeed has measure zero, just like the width of a point within a line has measure zero, it is better to just say co-dimension 1 and leave it at that. In addition, co-dimension 1 is true for proper LCS, yet FTLE ridges tend to be coarse (as you suggest in your Figure 2 and other parts of the paper) and therefore your "infinitesimal" description is confusing. Best to omit this part.*

This is a good point, and we agree that saying "co-dimension 1" is better phrasing than "infinitesimally thin". Furthermore, we agree that this only applies to LCS, whereas the FTLE is just a field of values, and as per the previous comment, we intend to make the distinction between FTLE and LCS more clear here as well.

*Line 152, you mention larger FTLE at initial time is due to a large velocity gradient, this suggests high FTLE is produced by the velocity horizontal shear in which case it is not an LCS. Also, if it is an LCS then attraction rather than accumulation would be better.*

Thank you for this remark. The note on a horizontal velocity shear is a good point, which we did not consider in our manuscript. While it is true that a horizontal velocity shear (shear dispersion) would yield large FTLE, this does not cause attraction/repulsion, and is therefore not an LCS. This brings us again back to the point that FTLE does not necessarily equate to LCS, which is a distinction that is lacking in our manuscript. We intend to discuss horizontal velocity shear in the revised manuscript.

*Line 160 you claim longer averaging periods effectively decrease variability. Although this makes sense intuitively, it is hard for me to see this by just looking at the figures. For example, there does not seem to be a large difference between the 7-day standard deviation and the 28-day one. Can you quantify this further?*

The idea here is that both the average and standard deviation appear to be concentrated with large values at particular locations (e.g. along the continental slope) for shorter time periods, but spreads out more throughout the domain for longer periods of time. Fig. 4 shows an example of this, whereas Fig. 9 intends to quantify this decrease in variability. We agree that this can be hard to see from Fig. 4, so this figure will be remade and we will consider presenting it alongside Fig. 9.

*Lines 167-168 can you discuss further the relation between a persistent current and persistent FTLE? Why is not surprising that they co-locate? Is it due to velocity shear?*

Thanks for the reminder; the text here will need some work. We are convinced that the underlying cause of both persistence and robustness (although we'll need to reconsider these terms) is the strong potential vorticity (PV) gradient set up by the steep continental slope. A steady current and recurring FTLEs are two of many results of this PV gradient. But, in relation to your cautionary point on shear dispersion, the velocity shear along the continental slope is probably a large contributor to FTLE formation here. This section will be expanded upon in a revised manuscript.

*Figure 6, some colors are saturated (especially e and f) so we can't get a sense of how large the values are, also the mean and the std deviation are not too far off, consider plotting them with the same scale (say 0.01 to 0.07 or whatever is needed so colors are not saturated over large regions). This will aid comparisons between mean and standard deviation.*

This will be fixed.

*172-173 It should be easy enough to test whether it is truly a transport barrier. How about releasing synthetic drifters on both sides of the barrier candidate and testing this directly? It would be nice to see results from individual members and some trajectory ensemble averages.*

This is a very nice suggestion which would aid our discussion. We have already started some Lagrangian particle calculations.

*Figure 9a, the legends for daily winter and summer seem the same color. Also, why do the spectra for ensemble members (c and d) seem to decrease monotonically from member 1 at the top to the last member on the bottom?*

The legends will be fixed. The spectral graphs are computed after first averaging over a set number of ensemble members. That is, the graph for "ensemble members = 1" is just showing the spectral variance from one ensemble member, but the graph for "ensemble members = 24" shows the spectral variance taken after first averaging all 24 ensemble members. The averaging effectively decreases variability over the domain. We realize that this section needs to be reworked as the exact method for obtaining the graphs is not clearly stated and confusing.

*Line 241 can you describe the dependence on the averaged members when only a few members are averaged?*

Here we pick the members at random, and there is a possibility that any member might deviate largely from the other members. When only picking a few members, e.g. four, there is a possibility that one of those might be largely different from the other three members. This can have a large impact on the calculated variability between these four members, and the results can differ if all four members are more similar. The impact on the results from one deviating member will be larger when few members are considered as opposed to when many members are considered.

*Line 242, it would be nice to see the three regions used for the spectra, as you mention, some circulation patterns are highly predictable for example circulation along a slope tends to be quite predictable along large portions of the slope.*

Good point. This will be included.

*Line 243 and 252, could it be that FTLE seems to be more robust than persistent due to your choice of T=1day? Short T should be expected to result in more transient features relative to longer integration times. See for example the papers cited in the comments above for lines starting at 35.*

This could definitely be the case, and we will look more into it. As stated previously, we are motivated by typical applications which are on these time scales. However, we intend to conduct a proper analysis of the integration time.

*Line 276, a repelling and attracting LCS cannot be parallel at the same location, if you go back to Dong et al you will see they describe an attracting LCS*

Thank you, this is an error on our side.

*277-278 and again we have the issue of T being relatively short at 1 day.*

This will be discussed (as outlined above).

*283 You mention other methods for detecting persistent LCS could yield more nuanced results. You also cite (line 343) a paper by Gouveia et al to say that large-scale features give rise to quasi-steady LCS. If you read the Gouveia et al paper carefully you will see they use a method to find quasi-steady LCS that was published in 2018 (https://doi.org/10.1038/s41598-018-23121-y). This later paper has over 40 citations according to Google Scholar, suggesting that many other studies are using that method to extract quasi-steady LCS. This topic is directly relevant to your study, so it seems your literature review is lacking. As you will see throughout my comments, the use of FTLE is a problematic issue that keeps coming up. The methodology published in 2018, and used by Gouveia et al, does not rely on FTLE, although there is some averaging. The difference in approach suggests a worthwhile discussion regarding the differences between that approach and your approach, including the strengths and weaknesses of each method.*

While other methods provide better means to identify LCS as linear features, our reason for using FTLE fields is to obtain a gridded spatial description of Lagrangian transport characteristics that can be analyzed using simple statistical methods, e.g. to calculate a mean and standard deviation of FTLE fields. In addition, we see the FTLE analysis as a practical tool for use in operational applications (e.g. contingency modeling), and we want to highlight the potential use of FTLE analysis in operational oceanography. Nevertheless, we agree that there are problems with the FTLE, and are familiar with better and more modern approaches for detecting LCSs. A revised manuscript will be more careful about the connection between LCS and FTLE, and provide a more detailed discussion on shortcomings of FTLE analysis as opposed to methods that identify LCS as linear features, such as quasi-steady LCS.

*288 it is also possible that a very strong FTLE shows up in the average, even if it does not persist much in time, especially if it recurs.*

This is a good point, we'll bring it up.

*358-359 Strong, persistent FTLE can be caused by persistent horizontal shear in which case it would not be indicative of LCS. Strong persistent FTLE can be expected in many coastal regions to be caused by horizontal shear. High FTLE next to the coastline, as in Fig. 5 during the summer or around 69.4N in the winter, for example, should be particularly suspect. You need to clarify throughout your paper that strong FTLE may be caused by horizontal shear, in which case FTLE is NOT indicative of an LCS, and mention that horizontal shear can be persistently high at certain locations such as a coastline. These locations, according to your suggestion, would have persistent LCS due to the persistent high FTLE, yet FTLE is not indicative of LCS if it is solely due to shear.*

Clearly a good point. As outlined above, we will be more careful about the distinction between FTLE and LCS, and specifically mention the case of shear dispersion. We may also attempt to identify regions (at least illustrate by an example) where the FTLE is produced by velocity shear.

*It is not enough to mention FTLE ridges only approximate LCS and to reference where to find the distinction between the two (line 144), because this distinction directly impacts the interpretation of our results, as has been explained above.*

We intend to make changes in the manuscript to clearly distinguish between FTLE and LCS. Furthermore, we will highlight issues with the FTLE as a tool for LCS detection and discuss other LCS detection methods from literature.

*382 "…by combining LCS analysis with ensemble prediction methods." The 2018 method to find quasi-steady LCS mentioned above should be discussed in this context as well. Can that method be used to find robust features in operational forecasting? Or is it complementary information to the ensemble methods you propose? Or are these two methods for differing purposes? Can you expand on how to use these methods to detect robust or persistent LCS in terms of operational oceanography? How concretely can these methods be applied? Can you suggest step-by-step instructions on how to implement these methods in an operational application? Can you give an example of how they have been used or can be used in operational oceanography? How feasible, useful, and accessible are these methods in operational oceanography?*

We will certainly expand on the possible operational uses/applications of FTLEs in an ensemble prediction system, possibly both in the introduction and conclusions. As for the 2018 method to find quasi-steady LCSs, we do not understand it well enough yet to be able to ascertain exactly how this method would respond to an ensemble average. However, as the 2018 paper handles time averages, we see the potential from an operational perspective of combining ensemble averaging with quasi-steady LCS to find robust features. We will not attempt to compute these in this study, but we intend to compare and discuss the two approaches in a revised manuscript. If a combined approach holds promise, we will strongly consider expanding on the use of ensemble models with quasi-steady LCS in a future study.

[Figure]

Fig. A: Backwards FTLE for different integration times (6, 12, 24, 48, 72, 168, and 672 hours). The final panel shows the monthly average of 24-hours long FTLE computations.

---

## Author Response (AR1)

**Answer to reviewer #1**

Thank you for your detailed comments, which highlight a number of fundamental inaccuracies as well as more subtle issues which we have attempted to address in our revised manuscript. We have been particularly conscious on multiple comments made by the reviewers, which pertain to the manuscript as a whole, and made substantial changes throughout the manuscript. In particular, we have clarified the differences between FTLE and LCS, and clearly stated our analysis applies to the FTLE analysis and the reason for why this method was chosen. A special attention was also given to the integration length in the FTLE analysis. Detailed answers for each review comment are given below:

**Referee comment:**

*1. The term "Lagrangian Coherent Structures" (henceforth LCSs) is used prominently in the title, and throughout the article. This term was coined by Haller in the early 2000s, and it is the continuing work of this group that the authors almost exclusively cite in this paper. However, the work of this group now very clearly defines LCSs in very specific ways, which are outlined clearly in Haller (Annu Rev Fluid Mech, 2015). These definitions, stated within the two-dimensional context in which the current paper is situated, relate to attempting to find curves towards/from there is extremal attraction/repulsion -- so-called "hyperbolic LCSs." It is these, and some allied entities, that are LCSs according to Haller and the group's definition, and so citing those papers and not using those definitions does not make sense. Moreover, Haller and collaborators in this and a range of other papers make clear that FTLEs are not necessarily these LCSs, and so prominently using the term LCSs in this paper is incongruous at best. Since the paper deals exclusively with FTLEs (and their averaging -- not even their ridges which some other authors may refer to as a type of LCS), the title and the paper should exclusively use the term FTLEs. Of course, one should position FTLEs in the context of LCSs -- which to a range of other authors represent an entire suite of techniques devised to extract coherent structures (based on differing definitions and intuition) from genuinely unsteady data. There are many recent reviews of this range of different methods for LCSs in this broader context in the literature: Balasuriya et al (Physica D, 2018), Hadjighasem et al (Chaos, 2017), Shadden (in: Transport and Mixing in Laminar Flows, Wiley, 2011, pp59). The authors are advised to consult these in positioning their work (as in line 85 when they say "Various methods have been proposed for LCS detection") and deciding whether the term LCS is actually appropriate.*

**Response:**

We agree that in this manuscript, we are dealing with the variability of FTLE fields which are not directly equivalent to Lagrangian Coherent Structures. Furthermore, we agree that there are some fundamental issues with the FTLE as a tool for LCS detection, and that more complete methods exist. We have clarified that in the revised manuscript, discussed the weaknesses of FTLE with regards to LCS, compared it to other existing methods and adapted the manuscript's title accordingly.

**Changes in manuscript:**

We have made changes to the manuscript to highlight that we are dealing with FTLE, and not LCS. LCS and their properties are mentioned in the text (), as well as mentioning that FTLE ridges may be used for LCS detection under certain conditions and briefly discuss the weaknesses of this method as an LCS detection method (lines). The manuscript now focuses on FTLE, and we reason our choice to use FTLE analysis as method to investigate statistical properties of flow features pertinent for operational applications (line 44–48)

**Referee comment:**

*2. The main conclusion seems to be that taking FLTEs and averaging them gives a better diagnostic of "robust" coherent structures. In doing such an averaging, there are some scientific issues related to time-parametrization and that I will come back to in a later point. However, I have some comments with respect to the issue of averaging FTLE fields to smooth out what the authors seem to think of as non-robustness, and thereby extracting robust structures. First, the FTLE fields are definitely associated with a time-of-flow, and since the authors use 24 hours exclusively here, they will be identifying local exponential stretching rates over the past 24 hours. If the authors were \*not\* interested in stretching over that time-scale but rather over, say, a one-month period, then rather than taking a 24 time-window, they should take a one-month window. This will definitely smear over ephemeral stretching. (By the way, when the authors use the term "ephemeral" this depends on the context. It seems that this means structures which do not persist over a longer time-scale, say a week or two. Within this time-frame, perhaps 24 hours is ephemeral -- but the authors seem to have decided to use 24 hours in the FTLE calculation by choice, and so it's to be expected that features related to 24-hour time scales are what will be revealed in the FTLE field.) Second, the discussion seems to indicate that the authors want to find stationary structures which persist over a longer time. In other words, to think of the velocity field as predominantly steady, with unsteady variations, and the goal would be to find structures which are stable with respect to the unsteadiness (which is assumed smaller). Well, if so, there are better ways to approach this. Rather than calculating FTLE fields from the full unsteady Eulerian data, they can obtain a dominant steady part of the Eulerian velocity. One way to do this would be to average the Eulerian data over an explicit time period -- this has the added advantage of being able to specify a time-scale (the time of averaging) over which the Eulerian field is assumed "mostly" steady, and hence one can ascribe a time-scale to one's conclusions. Another alternative would be to use some smoothening technique -- and again, if using something like a spatial filter associated with an explicit length-scale, one can ascribe a scale (a length-scale in this instance) of "accuracy" of the processed data to enable any conclusions to be stated in relation to that. These methods work directly on the Eulerian velocity field, rather than applying techniques such at FLTEs on data which to all intents and purposes (based on the conclusions reached) seem to be viewed as "noisy." Remove the noise first to avoid amplification of inaccuracies when doing additional computations. Thirdly, if stationary objects are sought, it seems that the authors want to look at the dominant steady component, and if one has a steady velocity field, Lagrangian methods are irrelevant. Lagrangian issues only make sense if there is unsteadiness, and one needs to follow the flow. If steady, simple techniques (drawing streamlines, Okubo-Weiss criterion, etc) on the Eulerian velocity field give perfectly good information on what's going on.*

**Response:**

We realize that the choice of time (the integration time) must be dicussed in the manuscript, and include an presentation of various choices for the integration time in the manuscript (Figure. 3). Furthermore, our objective is not to find structures which exist over longer time periods (e.g. longer than 24 hrs), but to identify predictable flow features. In our averaging in time over consecutive FTLE fields we sought to identify regions where high values in the FTLE appear *frequently*, while noting fact that individual structures will form, drift, and deform over various time scales. This has been clarified in the manuscript. We appreciate your suggestions on how we can approach this issue.

**Changes in manuscript:**
New analysis of the effect of the choice of integration time, and discussion on how this parameter should be chosen to fit the application in section 3.1.

**Referee comment:**

*3. Based on the conclusions that the authors seem to be reaching, there seems to be an incomplete understanding of what the FTLEs represent. The FTLE field $ \sigma_{t_0}^t $ represents a field at time $ t_0 $, associated with the stretching rate experienced by infinitesimal fluid parcels beginning at time $ t_0 $ and flowing until time $ t $. This rate is converted to an exponential one via the logarithm, and the time-of-flow $ T = |t-t_0| $ is used to time-average this quantity so that it's the average exponential rate over the time period considered. Note that there is absolutely no mention here of anything being a flow barrier, or a "coherent structure." Thus, the FLTE identifies regions in the time $ t_0 $ based on stretching rates over the time-of-flow (in this case, 24 hours in the past). When one time-averages the FTLE field, what exactly does that mean? Presumably (and this is not clearly stated), FTLE fields are generated for differing $ t_0 $s but the same time of flow (this is sometimes called time-windowing in the coherent structure community), but then are the authors averaging over $ t_0 $? (Give an explicit formula for the averaging, so that this is clear.) If so, they are taking scalar fields which are defined over different times $ t_0 $, each of which is associated with flow over a different time-window $ (t_0-T,t_0) $, and averaging them. If this is what is done, it needs to be explained clearly. But then, the interpretation of this needs to be carefully stated, since one is averaging over different initial times, and what one gets cannot be associated with a particular time instance (unlike one calculation of an FLTE field, which gives a field at time $ t_0 $). Of course, one might argue that calculationally the averaging tells us of how 24-hour motion calculated over several initial times (say 1 January to 31 January 2023) is averaged to give an "average exponential rate of motion in January 2023 when a time-scale of 24 hours is considered for the exponential rate". This would need to be explained, because it's quite awkward and hard to interpret.*

**Response:**

Motivated by this comment, we revised our interpretation of FTLE maxima as transport barriers. We instead expand the introductory section with more mention of previous works that investigate this relationship—and then return to the issue in the Discussion/Conclusions section. We are also more careful about how we define FTLE and how it describes the flow field. We now provide more details on our objective by applying time averaging, We

adopted several of the reviewer's ways of formulating these definitions and interpretations, and like to thank the reviewer for providing clarifications.

**Changes in manuscript:**

We avoid interpretation of high values in the FTLE field as strict transport barriers, and investigate whether these act as accumulation regions for particle transport. We have added equations showing how the FTLE averages are conducted to make the method description more concise (Eqs. 6 and 7).

**Referee comment:**

*4. The above point related to one aspect of time-parametrization: the $t_0$ in the field $\sigma_{t_0}^t$. Another time-parametrization issue is the $t$, and thus the time-of-flow. Everything in this paper has used a time-of-flow of 24 hours. Hence, everything is slaved to this time-scale -- the exponential rate is computed based on time-of-flow for this time-scale. This issue is buried in the paper, with the multitude of plots not mentioning this explicitly If a time-of-flow of 48 hours were chosen instead, how do things change? Basically, the calculation of an FTLE field explicitly picks out a time-scale, and this is not something which the authors have clarified. The results are explicitly associated with this time-scale, and no other. If the results are to be used in forecasting, why is this the correct time-scale? Or is this method robust to changing the time-scale? How does the time-scale interact with the time associated with the time-averaging as discussed in my previous point? Basically, the issue of TIME (initial plus time-of-flow in calculating the FTLE and the appropriate interpretation of the FTLE field, the times of computation chosen for averaging) is crucial, and needs to be carefully explained, interpreted, and robustness evaluated (if appropriate).*

**Response:**

The choice of 24 hours as integration period is not completely arbitrary but motivated by typical uses of ocean forecasting models. These are decision support tools for search-and-rescue operations, oil-spill modeling, ice-berg trajectory forecasts, and similar trajectory analysis which often require forecasts of a few hours up to a few days. But the comment is definitely warranted, and we now provide an analysis figure with examples of various integration lengths for the FTLE (time-of-flow). The figure indicates that our results are not overly sensitive to the integration period within integration length of 12 hours to 72 hours. But we do see some interesting differences. For example, for much longer integration periods of 15 to 30 days, we see that the FTLE analysis yields distinct linear features in low current velocity regions, especially in the deep basins off the continental slope In contrast, in the energetic flow regions over the shelf and slope, longer integrations tend to smear out features. Our interpretation is that the ability of the FTLE field to pick up LCSs (in a broad sense) depends on the integration period matching the time scale of the dynamics—which varies depending on the environmental conditions.

We also compare FTLE analysis based on long integration periods with the time-average of several short integration periods and see distinct differences in the two approaches. Whereas the time-average provides a description of regions that are typically abundant with FTLE ridges, the long-time integrations do not distinguish areas in the high-velocity region

but instead allow to better characterize low-velocity regions, which are less pertinent to time-critical contingency modeling.

We do not take an opinion on right or wrong time integration, but in the revised manuscript we discuss its impact on the analysis, and provide a view on how an appropriate time period may be selected depending on the application in focus.

**Changes in manuscript:**

As our study is motivated by typical ocean forecasting models, we have not changed the integration time in our results. We have been more clear about this motivation in the text (), and also provided a brief analysis of how the FTLE field changes based on the integration time and discussed the impact of this analysis on our results, included in section 3.1.

**Referee comment:**

*5. Returning to the definition of the FTLE mentioned earlier. It represents a field at time $t_0$, associated with the average exponential rate over flow from time $t_0$ to $t$. Note that there is absolutely no mention here of anything being a flow barrier, or a "coherent structure." Yes, there are papers in the literature which seem to indicate such a connection, but the reality is that it is unjustified. The early papers in this area used STEADY toy models which had saddle points with one-dimensional stable and unstable manifolds emanating from them, and since these manifolds are associated with exponential decay rates, came up with the idea that FTLE ridges had something to do with stable and unstable manifolds. And these manifolds were flow barriers in some way. However, this argument does not hold water, since there are examples such as in Haller (Physica D, 2011) which show that the stable/unstable manifold interpretation sometimes fails even in infinite-time flow. (And, getting back to a previous point related to hyperbolic LCSs, the fact that repelling/attracting do not necessarily occur as expected are also shown.) Real data is much worse: it is unsteady, and finite-time. Finite-time aspects are awkward for inferring exponential rates of growth, since any function over a finite-time can be bounded by an exponential. Unsteadiness is yet another problem, because (even in the infinite-time context) saddle points generalize to hyperbolic trajectories, and their stable/unstable manifolds move around (another reason why time-averaging is questionable). Furthermore, it is not clear what an "FTLE ridge" is -- one never gets a genuinely one-dimensional curve which is well-defined, but rather gets regions of larger FTLE values. Balasuriya et al (J Fluid Mech, 2016) provide an assessment and interpretation of what the FTLE means, with an emphasis on fluid motion, which helps understand these issues. In particular, for finite-time, unsteady data sets, using FTLEs and their ridges cannot easily reach conclusions regarding flow barriers and coherence. FTLEs explicitly look at exponential growth rates with respect to the time-of-flow considered, and that's about it. So when one takes FTLE fields, as done in this paper, and tries to reach conclusions regarding coherent structures such as eddies (as the authors do towards the end of the paper), this needs to be treated with suspicion, because it is not on any firm scientific grounds. The idea that eddies can be demarcated by FTLE ridges -- notwithstanding my earlier comments on finite-time and unsteadiness -- may go back to work in the 1990s (such as del-Castillo-Negrete, Knobloch, Pierrehumbert) who analyzed perturbed toy models. In these cases, the unperturbed models were explicit and steady, and had saddle points with stable and unstable manifolds. In some cases, the geometry was such that these manifolds encircled an eddy. Since the manifolds many be*

*discoverable using FTLE ridges (again subject to various caveats), in such cases ONLY, one might think of an eddy as being found using FTLE fields. However the interior of the eddy does NOT typically contain exponential stretching, and hence FTLE fields by themselves cannot be used to reliably identify eddies as the authors here are doing in their later figures. Actually, in the standard fluid-mechanical dichotomy between stretching and rotation, the eddies have the opposite of stretching, and thus the FTLE is exactly the wrong thing to use (Okubo-Weiss and related criteria may help; however as noted by many authors unsteadiness is a problem in using such Eulerian characteristics). Basically, statements such as "two additional regions are identified and considered as robust in Fig.7 " (line 211 in the paper) are, in this vein, questionable. Indeed, high FTLE values indicate greater separation, and hence LESS certainty!*

**Response:**

As outlined in relation to comment #1 and #2, we revised the manuscript to clarify that FTLEs and LCSs are not the same thing, and focus on the properties of the FTLE only as this is what we have investigated. We thank the reviewer for providing the paper by Balasuriya et al (J Fluid Mech, 2016), which we refer to when interpreting FTLEs*.* We use the term "FTLE ridges" loosely after a brief definition as "elongated areas of high FTLE values" (line 34-35) to discuss features formed by locally high FTLE values which can be seen in our figures of the FTLE field. However, we agree that we don't get a genuine and well-defined 1D curve in the FTLE field, and will make changes in the manuscript to reflect this, as our analysis is centered around a statistical analysis of FTLE fields to characteries their robustness and persistence.
We also agree that it is unjustified to identify eddies through the FTLE, which was notintended as the main focus in this manuscript, hence we removed this interpretation. We see that this point is not clear, and instead it seems that we are attempting to use the FTLE to identify eddies. In a revised manuscript, we do not see a need to focus on analysis of eddies.

**Changes in manuscript:**

Changes have been made to address the distinction between FTLE and LCS in the introduction and throughout the manuscript. Since we don't conduct any ridge detection in our manuscript, and the term "FTLE ridge" appears to be a common and well defined term in the scientific community, we have defined "FTLE ridge" as "elongated areas of high FTLE values" (line 35). Furthermore, we heed the advice from the reviewer about eddies. Eddies are important structures of the flow field, and their mention has not been completely removed from the manuscript, but we have made changes to the manuscript so that it does not seem like we are trying to identify eddies through FTLE

**Referee comment:**

*6. The impression given throughout is that the authors are examining robustness of LCSs. I've already talked about why it's actually FLTEs and not LCSs, but in this point I want to question whether robustness is the right thing. There are many recent papers which examine robustness of FTLE fields (Balasuriya, J Comp Dyn, 2020; Guo et al, IEEE Trans Visual Comp Graphics, 2016; Raben, Exp Fluids, 2014), but this paper is not one of them.*

*(There are a few other papers, from oceanographic situations, which the authors speak to in lines 339-344.) By "robustness," the authors seem to mean smearing over small time scale motion, in other words looking for entities which persist over longer times. This needs to be made clear throughout the manuscript. (I've talked about the time-parametrization issue previously; to smear over smaller time scales, one needs to simply choose appropriate times for the FTLE which are relevant to what one is looking for.) In Section 4.3, also, the word "uncertainty" is questionable because of this same reason -- the authors have no calculated any uncertainty (i.e., have not evaluated anything to do with uncertainty in the input data).*

**Response:**

We would like stress that our recommendation is a distinction between 'robustness' and 'persistence' of FTLE fields as calculated from ocean model forecasts. Much of the discussion in literature regards what we identify as persistence, i.e. time variability. In our discussion, robustness refers to similarities of FTLE features across an ensemble of model flow realizations. We are aware of the studies which address uncertainties in FTLE computations, and have included them in our manuscript. Our goal is to investigate this issue from an operational perspective and investigate how an FTLE field manifests itself in an ensemble prediction system, directly addressing uncertainty in forecasts.
Clarify this distinction and put less emphasis on time variability as this subject has been more extensively been addressed in other studies and needs no repetition. Furthermore, the reviewer suggested including an analysis of current velocities. We have chosen to not go into details in this analysis, but included figures showing velocity fields and their averages, as well as particle trajectories in different ensemble members.

**Changes in manuscript:**

The revised manuscript provides an early definition of robustness and persistence (lines 66-70), and what motivates us to investigate robustness. New figures 2 and 8 show particle trajectories. Figures 4, 5 and 6 now show time and ensemble averaged velocity fields. We point the reader to a detailed analysis of particle trajectories in line (110).

**Referee comment:**

*7. There are several statements around Equation (2) and (3) which are incorrect. The statements in line 101 are all incorrect: $\delta x$ is not the final location, but $x$ is, and the (1,1) term in Equation (2) then represents the partial derivatives of $x$ with respect to $x_0$, for example. The integral bounds in (3) are unclear and inconsistent with the previous equation. Presumably the authors mean something like $$x(t) = x_0 + \int_{t_0}^t u \left( x(\tau), t(\tau) \right) \mathrm{d} \tau$$, and then one needs to also clarify that $x(\tau), y(\tau)$ are the evolving trajectory locations. However, this integral formulation may not be the most natural. The differential form with $\dot{x} = u \left( x(t), y(t) \right)$, $\dot{y} = v \left( x(t), y(t) \right)$ and the initial condition $\left( x(t_0), y(t_0) \right) = \left( x_0, y_0 \right)$ connects better with the discussion, perhaps.*

**Response and changes in the manuscript**

Thank you for spotting the errors arround Eqs. 2 and 3. We have corrected the equations in section 2.3 according to the reviewers comments.

**Referee comment:**

*8. The discussion at the end of Section 2.3 is fraught. What is a "2D curve" [line 126]? The statement that "averaging smooths the ridges into fields of attraction/repulsion" [line 128] is incorrect because, as mentioned previously, claiming that FTLEs have anything to do with attraction/repulsion is questionable. "Making these ridges more certain" [line 130] relates to an earlier comment that what is being done here has nothing to do with robustness or certainty. The authors comment that the time- and ensemble-averaged FTLE fields are not transport barriers (which is correct -- but neither is the FTLE field -- and again it's not clear to me how a field can be a barrier), but then talk about things being barriers over larger regions. Figure 2 is strange. One does not usually get FTLE ridges (even in idealized steady toy models in 2D) which intersect and pile on each other like this. Intersections in such cases occur at saddle points. If unsteady, intersections that one gets, analogous to intersections between stable and unstable manifolds, can relate to chaotic motion -- but these are between forward-time FTLE ridges and backwards-time FTLE ridges, rather than self-intersections within one of these.*

**Response:**

We acknowledge that we have been too quick on equating high values in the FTLE field to attraction or repulsion, and that we used the term "transport barriers" too loosely. We have in fact only looked at the FTLE field, and not conducted any ridge detection or investigated further criteria from e.g. Farazmand and Haller (2012) to distinguish any potential LCS. Furthermore, as the second reviewer pointed out, strong values in the FTLE field may be produced by horizontal velocity shear, which does not result in attraction/repulsion. We have undertaken a major revision of the manuscrip with regard to interpretation of FTLEst, and avoid interpretation as "transport barriers". We do, however, believe there is such a thing as a dynamical transport barrier, specifically related to the strong potential vorticity (PV) gradient associated with a steep continental slope (as we have in our domain). A PV barrier is not impenetrable, but it does inhibit tracers and particles - as documented throughout the dynamical literature (we will add references). And we do believe that the high FTLE values over the continental slope, also when averaging over time or over the ensemble, do in fact arise from the strong PV gradient (or 'PV barrier' if one allows a more loose use of the term). n the revised manuscript, we will be more careful in explaining how we use the term, aware that different parts of the research community might be using various degrees of rigor in their definitions.

We agree that Fig. 2 is misleading and have removed it from the manuscript. Instead, we present examples of FTLE ensemble realisations form the ocean model along with the ensemble average and a time average around this situation in Fig. 8.

**Changes in manuscript:**

We have made it more clear that FTLE is not the same as LCS, and focused on discussing FTLEs. We also added a discussion about potential vorticity and FTLEs produced by horizontal velocity shear. Fig. 2 has been removed form the manuscript. New Figure 8 shows example of ensemble realisations of FTLE with particle positions along with ensemble average and time average.

**Referee comment:**

*9. The fact that a large standard derivation of the time-averaged FTLE usually is close to where the FTLE is large [line 169-171] is no surprise. Large FTLE relates to larger uncertainty in the results, because any errors (based on interpolation to a grid, say) increase exponentially. Hence, the values one assigns to the FTLE tend to be less certain. This issue is well-known, and described in some of the papers I've mentioned previously on robustness of FTLEs.*

**Response:**

Thank you for this remark. This will be considered when reworking the manuscript.

**Changes in manuscript:**

After closer consideration, we decided that the standard deviation did not add much to our analysis, and have removed all figures of standard deviation.

**Referee comment:**

*10. For the discussion on Section 3.4, I can't quite understand what the $ k $ in the figures is. Since in 2D, one has two wavenumbers -- say $ l $ and $ m $, associated with the eastward and northwards coordinates. I don't understand the discussion [lines 230 onwards] about averaging over rows and columns (over $ l $ and $ m $?). Is $ k = \sqrt{l^2 + m^2} $? The spectral plots in Figure 9 are used to infer robustness in some way, based on the fact that one gets decay with $ k $ in the bottom figures, say. Any smoothening process will of course get rid of the smaller wavenumbers. Similarly one expects more smoothness when averaged over more and more days.*

**Response:**

The procedure for producing Fig. 9 has been poorly described in the manuscript. The transformation of the FTLE field to Fourier space has been done using the 2D discrete cosine transformation method. As pointed out, this results in two wavenumbers, m and l. These are combined to create a radial wavenumber k = sqrt(m^2 + l^2). We intend to revise this section, include equations and references, and clarify it in the discussion.

**Changes in manuscript:**

Equations and references are included in section 3.4. Text in the section has been revised to explain how what we do and how we obtain the Fig. 9 (now Fig. 7) more clearly.

**Answer to reviewer #2**

Thank you for your detailed comments, which highlight a number of fundamental inaccuracies as well as more subtle issues which we have attempted to address in our revised manuscript. We have been particularly conscious on multiple comments made by the reviewers, which pertain to the manuscript as a whole, and made substantial changes throughout the manuscript. In particular, we have clarified the differences between FTLE and LCS, and clearly stated our analysis applies to the FTLE analysis and the reason for why this method was chosen. However, we realize that the quasi-steady LCS method might be more proper, and while we did not compute them here, we have given the method more attention in the discussion. A special attention was also given to the integration length in the FTLE analysis. Detailed answers for each review comment are given below:

**Referee comment:**

*20 I would suggest you emphasize the sensitivity of trajectories and not the sensitivity of the velocity because even with a hypothetically perfect velocity, a small error in a trajectory's initial position can grow exponentially into large errors. The idea of your paper is to introduce perturbations in the velocity and measure how trajectories respond. Note the interest is in the trajectory response as visualized through LCS given a velocity ensemble spread. The main concern is trajectory uncertainty, even if explored in terms of velocity uncertainty.*

**Response:**

This is a good point and we will adapt this change in wording.

**Changes in manuscript:**

Wording has been adapted (lines 50-53).

**Referee comment:**

*In 35 you mention:*

*Previous studies often discuss the LCS methodology and their practical applications, but rarely touch upon the topic of LCS estimates being inherently affected by uncertainties in the velocity fields they aim to describe. Furthermore, short-lived flow features constantly develop, drift, and dissipate in real oceanic flow (Chen and Han, 2019). Given their time-dependency, LCSs might appear and disappear just as quickly. This brings up two important questions: (1) Given the velocity field uncertainty, how robust, i.e. predictable, are LCSs derived from ocean models at a particular time?; (2) Given their time-dependency, how persistent are LCS in ephemeral flows?*

*I don't think robust can be equated with predictable. Robust in your study means that different realizations of a simulation (i.e. similar simulations) result in the same LCS. There is no predictive capability (i.e. estimates of future information based on past information) in this analysis*

**Response:**

We agree that predictability does not directly follow from robustness. This is only be the case if the underlying circulation model is proven to have both predictive skill on both the time and the spatial scales of interest, and if the ensemble spread exhibits reliability. Both criteria are valid to a limited extent in terms of surface currents, which has been investigated by Idzanovic et al. (2023) for the model system that is used here. Our study is interested in whether the EPS yields similar FTLE fields at a specific time, and does not focus on their predictability. However, when using the method in concrete applications one must confirm that the underlying EPS has reliability in the ensemble spread for surface currents. This point is clarified in the manuscript.

**Changes in manuscript:**

Wording has been changed in the manuscript and this point has been clarified. A statement on the requirement of reliability has been added to section 4.5 §3.

**Referee comment:**

*The short-lived structures you mention are not a problem, or even interesting, as it is straightforward to filter them and find the prominent deformation patterns, without the need to average see e.g. Olascoaga & Haller (2012; https://doi.org/10.1073/pnas.111857410) or Kunz et al (https://doi.org/10.5194/egusphere-2024-1215) that discusses the importance of persistence when it comes to attracting hyperbolic patterns and the lack of meaningful influence with short-lived structures (in particular while hyperbolic structures are forming or decaying). These are results without ensembles or any type of averaging. In particular, Olascoaga & Haller (2012) get rid of short-lived LCS by increasing the integration time T to 15 days. Thus, the choice of T= 1 day in your study raises the question of how do your results depend on your choice of T? Are the transient FTLE features that you filter through averaging unnecessarily increased by this choice? Wouldn't it be better to use a longer integration time to filter those features instead of time averaging? Also, as I will mention below, there are several papers showing 1) how to find persistent (or quasi-steady) LCS and 2) that persistent LCS are ubiquitous and meaningful. It is true however that we do want to be able to discern which are short-lived and not meaningful, and there is more than one way to get there.*

**Response:**

Reviewer #1 raised a similar concern about the integration period. The choice of 24 hours as integration period is not completely arbitrary but motivated by typical uses of ocean forecasting models. These are decision support tools for search-and-rescue operations, oil-spill modeling, ice-berg trajectory forecasts, and similar trajectory analysis which often require forecasts of a few hours up to a few days. But the comment is definitely warranted, and now provide an analysis figure with examples of various integration lengths for the FTLE (time-of-flow). The figure indicates that our results are not overly sensitive to the integration period within integration length of 12 hours to 72 hours. But we do see some interesting differences. For example, for much longer integration periods of 15 to 30 days, we see that the FTLE analysis yields distinct linear features in low current velocity regions, especially in the deep basins off the continental slope In contrast, in the energetic flow regions over the shelf and slope, longer integrations tend to smear out features. Our interpretation is that the

ability of the FTLE field to pick up LCSs (in a broad sense) depends on the integration period matching the time scale of the dynamics—which varies depending on the environmental conditions.

We also compare FTLE analysis based on long integration periods with the time-average of several short integration periods and see distinct differences in the two approaches. Whereas the time-average provides a description of regions that are typically abundant with FTLE ridges, the long-time integrations do not distinguish areas in the high-velocity region but instead allow to better characterize low-velocity regions, which are less pertinent to time-critical contingency modeling.

We do not take an opinion on right or wrong time integration, discuss the impact of this analysis in the revised manuscript, and provide a view on how an appropriate time period may be selected depending on the application in focus.

**Changes in manuscript:**

As our study is motivated by typical applications in operational oceanography, we have not changed the integration time in our results. We have been more clear about this motivation in the text (lines 143-147), and also provided a brief analysis of how the FTLE field changes based on the integration time and discussed the impact of this analysis on our results. This analysis added into section 3.1.

**Referee comment:**

*Schematic 2 is not correct, the average of any number of zeros is still zero, i.e. regions where FTLE is zero in the left side of the schematic should also be zero on the right. Notice that in the caption of Figure 2 you introduce a concept that is not mentioned, or used, in any other part of the paper and that is "the average region covered by them". There must be a better way to convey the idea you want to convey.*

**Response:**

We agree that Fig. 2 is misleading and that structures like these do not appear in nature. This is also a concern that was raised by reviewer #1. The idea behind Fig. 2 was to show what FTLEs might look like over the ensemble dimension, where each drawn line represents FTLE ridges from different ensemble members. This is not shown clearly enough in neither the figure nor the text, and has been removed.

**Changes in manuscript:**

Fig. 2 has been removed. New Figure 8 shows example of ensemble realisations of FTLE with particle positions along with ensemble average and time average.

**Referee comment:**

*40 should be Gulf Stream*

**Response:**

This will be fixed.

**Changes in manuscript:**

Fixed.

**Referee comment:**

*40 The paper by Badza et al 2023 does not present reliable results (this is a paper that should have been rejected in my view) because:*

*They use a stochastic differential equation for the velocity to measure the robustness of LCS methods to noise. This is a big mistake. The mathematical theory of variational LCS explicitly states the results are only valid for a deterministic velocity, i.e. the results only hold for the typical ordinary differential equation dx/dt = v(x(t),t). This is a very basic, yet fundamental mistake that renders their results meaningless.*

*Even if the theory of hyperbolic LCS were to hold for a stochastic vector field, a stochastic component is not representative of the uncertainty typically encountered in a geophysical velocity field, neither simulated nor observed. An ensemble of simulations is a much better choice for uncertainty.*

*In their Gulf Stream case, they allow for a very long integration time (three months) while computing LCS within a limited spatial region, Thus the results are plagued by fictitious boundary effects, as is evident from their figures. A simple computation shows the inadequacy of their choice: Their domain is 30 degrees wide (note the flow is mainly west to east in their domain). That means their domain is less than 30\*111=3330 km wide. Yet their integration time is 90 days, which means you would only need a velocity of 37 km/day (0.43 m/s) to traverse the whole domain, from west to east. The Gulf Stream commonly reaches a velocity of over 150 km/day (above 1.5 m/s). Indeed, boundary effects in their results are apparent, and they mention it themselves: "Most of these streaks appear to look like diagonal lines, which is likely attributable again to the exodus of particles over the large period of flow considered." They also mention in their discussion that: "As with most of the previously discussed methods, this can be attributed to the exodus of particles from the domain over our 90 day flow period." Even without the two mistakes mentioned above, there is not much that can be learned from results plagued by unphysical boundary effects.*

*In your study, Badwa et al. 2023 are cited to say that hyperbolic LCS detection is not reliable. However, as explained above their conclusion is meaningless. I therefore, as a reviewer, make the extraordinary suggestion that you delete, or at least adequately discuss, any sentence citing the Badza et al. paper, to avoid amplifying misleading results. The other papers you cite such as Harrison & Glatzmaier are better and are adequate for the point you wish to make regarding FTLE. In particular, the representation of uncertainty they choose is realistic and does not involve a fundamental dynamical-systems mistake.*

**Response:**

Thank you for this comment. We did not realize this issue with Badza et al. (2023), and removed the citation from our manuscript.

**Changes in manuscript:**

The Badza el at. (2023) citation and discussion around it has been removed from our manuscript. We instead conducted more literature research and found other articles to reference to.

**Referee comment:**

*55 what do you mean by dynamically active shelf region? Is there such a thing as a dynamically inactive sea? Either explain clearly the idea you want to convey or delete statements that don't add useful information, yet leave the reader wondering.*

**Response:**

What we meant here is that the shelf and slope region has more variability due to the presence of a complex coastline and the juxtaposition of different water masses (setting up frontal zones) that generate various forms of flow instability. We have adapted the wording in the manuscript.

**Changes in manuscript:**

This has been fixed (line 71).

**Referee comment:**

*In the caption of Figure 1 you mention Mockstraumen has been indicated by an arrow, consider mentioning in the text what is this region. why is it important?*

**Response:**

The velocity field through Mockstraumen is highly tidally dependent, and we investigated whether this will yield periodic and thus predictable FTLE fields. Figures of this are not included in the manuscript, but are mentioned in the text (line 207-211 ).

**Changes in manuscript:**

We now mention Mockstraumen in lines 89-90.

**Referee comment:**

*100 Although it is true that fluid parcels need to be advected, Equation 3 is not an accurate description of how Equation 2 is computed. It needs to be clear that you are time-integrating the velocity along a path which is not the same as integrating the velocity with respect to time at a fixed location, as your equation suggests. Importantly, you need to integrate two trajectories to be able to compute the distance \partial x, and it is not enough to just integrate the velocity as your equation reads.*

**Response:**

Thank you for pointing this out. We have updated the equation and text.

**Changes in manuscript:**

The Eq. 3 (now Eq. 1) and corresponding text has been updated and moved to the beginning of section 2.3. Lines (116-121)

**Referee comment:**

*105 Although deformation is indeed given by the singular values of the Jacobian of a Flow map, there is no such information as a "speed of deformation" embedded in the Flow map (note speed has units distance/time).*

**Response:**

This will be fixed.

**Changes in manuscript:**

Fixed in line 135.

**Referee comment:**

*110 there is nothing to show, that is the definition of FTLE.*

**Response:**

This will be fixed.

**Changes in manuscript:**

Fixed in line 137.

**Referee comment:**

*115 "Largest FTLE = LCS" is not true. This needs to be explained in detail throughout the paper so that your conclusions are not misleading. See my comments about strong FTLE produced by large horizontal shear in coastal regions in what follows.*

**Response:**

The other reviewer also commented on this. We realize that the largest FTLE does not necessarily imply an LCS and that there are additional criteria which an FTLE ridge must satisfy to qualify as an LCS approximation. We have clarified this difference between FTLE and LCS in the introductory section, and made remarks about the weaknesses of FTLE.

**Changes in manuscript:**

We have been more mindful about equating FTLE to LCS, and made the distinction between these two more clear. We have made changes throughout the manuscript reflecting that we are working with FTLE and not LCS throughout the manuscript. Weaknesses of the FTLE are mentioned in lines 43-45 and discussed section 4.1.

**Referee comment:**

*125 "infinitesimally thin" is an unusual description. Although the width of a line within a plane indeed has measure zero, just like the width of a point within a line has measure zero, it is better to just say co-dimension 1 and leave it at that. In addition, co-dimension 1 is true for proper LCS, yet FTLE ridges tend to be coarse (as you suggest in your Figure 2 and other parts of the paper) and therefore your "infinitesimal" description is confusing. Best to omit this part.*

**Response:**

This is a good point, and we agree that saying "co-dimension 1" is better phrasing than "infinitesimally thin". Furthermore, we agree that this only applies to LCS, whereas the FTLE is just a field of values, and as per the previous comment, we intend to make the distinction between FTLE and LCS more clear here as well.

**Changes in manuscript:**

Section 2.3 now focuses directly on FTLE instead of FTLE as a tool for LCS detection.

**Referee comment:**

*Line 152, you mention larger FTLE at initial time is due to a large velocity gradient, this suggests high FTLE is produced by the velocity horizontal shear in which case it is not an LCS. Also, if it is an LCS then attraction rather than accumulation would be better.*

**Response:**

Thank you for this remark. The note on a horizontal velocity shear is a good point, which we did not consider in our manuscript. While it is true that a horizontal velocity shear (shear dispersion) would yield large FTLE, this does not cause attraction/repulsion, and is therefore not an LCS. This brings us again back to the point that FTLE does not necessarily equate to LCS, which is a distinction that was lacking in our manuscript. We now discuss this in the revised manuscript.

**Changes in manuscript:**

Discussions about FTLE produced by horizontal velocity shear are now included in lines 43-45, 206, 287-299 and 252.

**Referee comment:**

*Line 160 you claim longer averaging periods effectively decrease variability. Although this makes sense intuitively, it is hard for me to see this by just looking at the figures. For example, there does not seem to be a large difference between the 7-day standard deviation and the 28-day one. Can you quantify this further?*

**Response:**

The idea here is that both the average and standard deviation appear to be concentrated with large values at particular locations (e.g. along the continental slope) for shorter time periods, but spreads out more throughout the domain for longer periods of time. Fig. 4

shows an example of this, whereas Fig. 9 intends to quantify this decrease in variability. We agree that this can be hard to see from Fig. 4, so this figure will be remade and we will consider presenting it alongside Fig. 9.

The idea here was the both the average and standard deviation appeared to be concentrated with large values at particular locations (e.g. along the continental slope) for shorter time periods, but spread out more throughout the domain for longer periods of time. Fig. 4 showed an example of this, whereas Fig. 9 intended to quantify this decrease in variability. However, we did not feel that the short term average or the standard deviation added much value to our analysis, and have been removed.

**Changes in manuscript:**

Fig. 4 has been replaced by the new Fig. 4. After consideration, we decided that the standard deviation did not add much to the analysis, and figures of standard deviation have been removed. Furthermore, we no longer show time averages for shorter time periods, but kept Fig. 9 (now Fig. 7) to quantify the variability.

**Referee comment:**

*Lines 167-168 can you discuss further the relation between a persistent current and persistent FTLE? Why is not surprising that they co-locate? Is it due to velocity shear?*

**Response:**

Thanks for the reminder. We are convinced that the underlying cause of both persistence and robustness is the strong potential vorticity (PV) gradient set up by the steep continental slope. A steady current and recurring FTLEs are two of many results of this PV gradient. But, in relation to your cautionary point on shear dispersion, the velocity shear along the continental slope is probably a large contributor to FTLE formation here.

**Changes in manuscript:**

We added a discussion about the persistent PV gradient set up by the steep continental slope in section 4.1, where we argue that this will have an effect on particle transport. Here, we discuss that this PV gradient likely is one of the reasons that persistent FTLEs are seen along the continental slope (lines 294-300). Velocity shear is also mentioned in lines 43-45, 206, 287-299 and 252.

**Referee comment:**

*Figure 6, some colors are saturated (especially e and f) so we can't get a sense of how large the values are, also the mean and the std deviation are not too far off, consider plotting them with the same scale (say 0.01 to 0.07 or whatever is needed so colors are not saturated over large regions). This will aid comparisons between mean and standard deviation.*

**Response:**

We decided that the standard deviation did not add much value to our discussion.

**Changes in manuscript:**

This Fig. 6 has been replaced by the new Fig. 6.

**Referee comment:**

*172-173 It should be easy enough to test whether it is truly a transport barrier. How about releasing synthetic drifters on both sides of the barrier candidate and testing this directly? It would be nice to see results from individual members and some trajectory ensemble averages.*

**Response:**

This is a very nice suggestion which would aid our discussion. We started Lagrangian particle calculations for this purpose, but ended up not including results indicating transport barriers. Instead we include references where this experiment has been conducted. We did however add a Figs. 2 and 8, showing an example of particle trajectories from individual ensemble members.

**Changes in manuscript:**

We added Fig. 2, which shows a simple example of how particles are advected differently in different ensemble members, and point the reader to de Aguiar et al (2023) (line 110 ), where the authors conduct a detailed analysis of particle advection in different Barents 2.5 EPS ensemble members. Also added Fig. 8 which shows a case where particles from different ensemble realizations are attracted to an ensemble averaged FTLE feature.

**Referee comment:**

*Figure 9a, the legends for daily winter and summer seem the same color. Also, why do the spectra for ensemble members (c and d) seem to decrease monotonically from member 1 at the top to the last member on the bottom?*

**Response:**

The spectral graphs are computed after first averaging over a set number of ensemble members. That is, the graph for "ensemble members = 1" is just showing the spectral variance from one ensemble member, but the graph for "ensemble members = 24" shows the spectral variance taken after first averaging all 24 ensemble members. The averaging effectively decreases variability over the domain.

**Changes in manuscript:**

The legends have been fixed. We expanded upon the method used to create Fig. 9 (now Fig. 7), and the text in section 3.4 is now more clear.

**Referee comment:**

*Line 241 can you describe the dependence on the averaged members when only a few members are averaged?*

**Response:**

Here we pick the members at random, and there is a possibility that any member might deviate largely from the other members. When only picking a few members, e.g. four, there is a possibility that one of those might be largely different from the other three members. This can have a large impact on the calculated variability between these four members, and the results can differ if all four members are more similar. The impact on the results from one deviating member will be larger when few members are considered as opposed to when many members are considered.

**Changes in manuscript:**

We added a note about the dependency of the ensemble average on the specific members included for few members in lines 258-260.

**Referee comment:**

*Line 242, it would be nice to see the three regions used for the spectra, as you mention, some circulation patterns are highly predictable for example circulation along a slope tends to be quite predictable along large portions of the slope.*

**Response:**

Good point. This has been included.

**Changes in manuscript:**

A figure showing the three regions used for the computation has been added to Fig. 7.

**Referee comment:**

*Line 243 and 252, could it be that FTLE seems to be more robust than persistent due to your choice of T=1day? Short T should be expected to result in more transient features relative to longer integration times. See for example the papers cited in the comments above for lines starting at 35.*

**Response:**

This could definitely be the case, and could be investigated more. As stated previously, we are motivated by typical applications which are on these time scales. However, we have now added a brief analysis of the integration time.

**Changes in manuscript:**

We ended up not pursuing this topic into more detail, but have made it clear that we are motivated by short-term applications which motivates our choice of T. An analysis of the integration time has been included in section 3.1.

**Referee comment:**

*Line 276, a repelling and attracting LCS cannot be parallel at the same location, if you go back to Dong et al you will see they describe an attracting LCS*

**Response:**

Thank you, this was an error on our side.

**Changes in manuscript:**

This has been fixed.

**Referee comment:**

*277-278 and again we have the issue of T being relatively short at 1 day.*

**Response:**

This has been discussed (as outlined above).

**Changes in manuscript:**

A discussion around the choice of T has been added in section 3.1.

**Referee comment:**

*283 You mention other methods for detecting persistent LCS could yield more nuanced results. You also cite (line 343) a paper by Gouveia et al to say that large-scale features give rise to quasi-steady LCS. If you read the Gouveia et al paper carefully you will see they use a method to find quasi-steady LCS that was published in 2018 (https://doi.org/10.1038/s41598-018-23121-y). This later paper has over 40 citations according to Google Scholar, suggesting that many other studies are using that method to extract quasi-steady LCS. This topic is directly relevant to your study, so it seems your literature review is lacking. As you will see throughout my comments, the use of FTLE is a problematic issue that keeps coming up. The methodology published in 2018, and used by Gouveia et al, does not rely on FTLE, although there is some averaging. The difference in approach suggests a worthwhile discussion regarding the differences between that approach and your approach, including the strengths and weaknesses of each method.*

**Response:**

While other methods provide better means to identify LCS as linear features, our reason for using FTLE fields is to obtain a gridded spatial description of Lagrangian transport characteristics that can be analyzed using simple statistical methods, e.g. to calculate a mean and standard deviation of FTLE fields. In addition, we see the FTLE analysis as a practical tool for use in operational applications (e.g. contingency modeling), and we want to highlight the potential use of FTLE analysis in operational oceanography. Nevertheless, we agree that there are problems with the FTLE, and are familiar with better and more modern approaches for detecting LCSs. We have been more careful about the connection between LCS and FTLE, and provide a more detailed discussion on shortcomings of FTLE analysis as opposed to methods that identify LCS as linear features, such as quasi-steady LCS.

**Changes in manuscript:**

We now argue why we use FTLE in the introduction (lines 45-48), but try to highlight the problems of FTLE as an LCS detection tool (lines 43-45). The manuscript now focuses on FTLE themselves instead of LCS detected from FTLE.

**Referee comment:**

*288 it is also possible that a very strong FTLE shows up in the average, even if it does not persist much in time, especially if it recurs.*

**Response:**

This is a good point..

**Changes in manuscript:**

We added a note about this in lines 330-331.

**Referee comment:**

*358-359 Strong, persistent FTLE can be caused by persistent horizontal shear in which case it would not be indicative of LCS. Strong persistent FTLE can be expected in many coastal regions to be caused by horizontal shear. High FTLE next to the coastline, as in Fig. 5 during the summer or around 69.4N in the winter, for example, should be particularly suspect.You need to clarify throughout your paper that strong FTLE may be caused by horizontal shear, in which case FTLE is NOT indicative of an LCS, and mention that horizontal shear can be persistently high at certain locations such as a coastline. These locations, according to your suggestion, would have persistent LCS due to the persistent high FTLE, yet FTLE is not indicative of LCS if it is solely due to shear.*

**Response:**

Clearly a good point. As outlined above, we have been more careful about the distinction between FTLE and LCS, and specifically mention the case of shear dispersion.

**Changes in manuscript:**

A discussion about horizontal velocity shear has been added as outlined above.

**Referee comment:**

*It is not enough to mention FTLE ridges only approximate LCS and to reference where to find the distinction between the two (line 144), because this distinction directly impacts the interpretation of our results, as has been explained above.*

**Response and changes in manuscript:**

We have made changes in the manuscript to clearly distinguish between FTLE and LCS. Furthermore, we have highlighted issues with the FTLE as a tool for LCS detection and discuss other LCS detection methods from literature.

**Referee comment:**

*382 "…by combining LCS analysis with ensemble prediction methods." The 2018 method to find quasi-steady LCS mentioned above should be discussed in this context as well. Can that method be used to find robust features in operational forecasting? Or is it complementary information to the ensemble methods you propose? Or are these two methods for differing purposes? Can you expand on how to use these methods to detect robust or persistent LCS in terms of operational oceanography? How concretely can these methods be applied? Can you suggest step-by-step instructions on how to implement these methods in an operational application? Can you give an example of how they have been used or can be used in operational oceanography? How feasible, useful, and accessible are these methods in operational oceanography?*

**Response:**

We have expand on the possible operational uses/applications of FTLEs in an ensemble prediction system. As for the 2018 method to find quasi-steady LCSs, we do not understand it well enough yet to be able to ascertain exactly how this method would respond to an ensemble average. However, as the 2018 paper handles time averages, we see the potential from an operational perspective of combining ensemble averaging with quasi-steady LCS to find robust features. We have not attempted to compute these in this study, but have discussed the two approaches in a revised manuscript.

**Changes in manuscript:**

We have not computed quasi-steady LCS, but seeing as this is a more modern method for LCS detection, we added a discussion about quasi-steady LCS and their ability to detect persistent features. We also expand on the idea that because an averaging is conducted when finding persistent quasi-steady LCS in Gouveia et al (2020), a similar average over ensemble members can be conducted to find robust quasi-steady LCS. This is included in section 4.1.

---

## Author Response (AR2)

*The paper is much improved, in my opinion, and the equations are now clearer. The authors have carefully addressed reviewer comments. In general, I dislike requesting additional work after a second round of reviews, but I do believe some issues need addressing before your paper can conclude accurately on the persistence of FTLE. I also think that with a bit more work, your paper will be a better contribution to our understanding of the robustness and persistence of FTLE/LCS. As it is right now, I feel your conclusions could be misleading (due to overlooking a possibility I explain below). I do believe my suggestions will represent a moderate amount of work.*

*Some other issues could (and should) have been corrected through a more careful review by the authors and by using free software for grammar/spelling checking. I begin with the main issue.*

*Main issue:*
*In Figure 7b, variance diminishes as the averaging period increases. Your interpretation is that: "Meanwhile, the spatial variance of Ft continues diminishing, signifying that FTLE fields are more robust over many members than persistent over long time periods." You also mention: "The decay of variance as more FTLE fields are included in an average (Fig. 7a vs. Fig. 7b) indicates that FTLE features are more robust than persistent." However, figure 8h gives us a clue of what could be happening: FTLE evolves over time by changing positions (as one would expect in an unsteady flow) so that the time-average of these FTLE results in a more homogenous field (less variance). In other words, if we were to look at the FTLE fields that are averaged, we would likely see persistent FTLE features that, over time, move slightly in space, e.g., by meandering and other types of small changes in location. This is especially true at locations with a slope that tends to anchor the mean flow. The average would then capture a smeared field. Instead of distinct FTLE ridges localized at one location (as in 8g), we see diluted/smeared FTLE over a greater area. However, does a transport barrier that meanders not prevent trajectories from crossing the barrier just because it meanders? Can we say that a transport barrier that meanders is not persistent just because there is a temporal evolution to the exact location of the barrier, even if the meandering is relatively small, and it happens about a fixed location? I feel there is a strong possibility that time averaging LCS/FTLE is not the best (or only) way to determine persistence due to the nature of unsteady geophysical flows. However, the work you have done is still important to understand this. I believe it will be simple enough to assess persistence in a different way: first, ensemble-average the velocity at each time, then compute FTLE using the ensemble-averaged velocity, and finally, time average the FTLE over periods of increasing length as you already did. Presumably, the ensemble averaging of the velocity will remove the meanders and leave behind the persistent flow features. Now assess the persistence of the FTLE without being thrown off by small changes in location. I am not advocating for one method over the other. All I am saying is that without testing both methods, there is a high likelihood that we are left with an incomplete picture and that conclusions solely based on the incomplete picture may be misleading. After careful thought, I do believe that the topic of LCS persistence is a nuanced one. By comparing both methods of time-averaging FTLE, we will gain better insight into their persistence, and the user can decide which method suits best their needs. The method you tested is: you compute FTLE from each ensemble member, then ensemble-average FTLE, and then average FTLE in time over increasing windows. The method I suggest is: first, ensemble-average the velocity, then compute FTLE, and then time average FTLE over increasing windows. I do believe that with these additions,*

*your paper will be a better contribution to our understanding of the robustness and persistence of LCS/FTLE. I also fear that your current conclusions that FTLE is not as persistent may be misleading, as explained above with an example based on meandering. You do seem to suggest that first averaging the velocity is not a good idea (although it is unclear what type of averaging you refer to, more on that later). But I argue that without first computing the ensemble average velocity, and then estimating persistence, we are left wondering what exactly is the meaning of the lack of persistence that your report. It could be an efficient barrier persisting in time but not showing up in the averages except as spatially smeared, due to small spatial excursions over time. I argue we need both types of time averages to discern, and truly understand.*

This is a very nice suggestion. In fact, similar arguments crossed our mind when we initially started working on the manuscript, but we decided then that we wanted to keep the time and ensemble dimensions separate. However, after taking the reviewer comment into account, we think this is a very good idea that should be investigated further.

We have therefore computed FTLE fields following the reviewer's suggestion, that is by ensemble-averaging velocity fields first and then examining time averages over the resulting FTLE fields. A paragraph describing this approach is added in lines 166-172. A new figure has been added (Fig. 7) which shows such time averages over December, January and February. Curves showing how the spatial variance of the new FTLE fields respond to time averaging have also been added to Fig. 8b. Text describing these new results is added in lines 252-261, 275-276 and 356-359.

As suggested by the reviewer, ensemble-averaging the velocity field removes the most unpredictable and time variable flow features. Thus the resulting FTLE fields contain less chaotic features, and the resulting time-averaged FTLE fields are less "noisy", and highlight ridges which are likely to be persistent.

*Other issues:*

*In the abstract, you mention:*
*"Generally, Lagrangian trajectories as well as FTLE analysis inherit uncertainty from the underlying ocean model, bearing substantial uncertainties as a result of chaotic and turbulent flow fields."*
*Trajectories can indeed suffer from large uncertainty in different realizations of a velocity. Even a small spatial difference in the location of a repelling LCS will cause a large difference for a trajectory initiated at the same location and time (you can see an example of this in the Wikipedia entry for Lagrangian Coherent Structure where they compare a real saddle-point type structure with the same structure from a model that places the structure slightly shifted from the real location). However, many of us have noted that LCS tend to be robust to small changes in the velocity field (with no "substantial uncertainties"), and this paper aims to investigate the robustness of LCS. Thus, I believe you are stating that FTLE inherits substantial uncertainty without much support. Indeed, later in the abstract, you mention that you find FTLE to be robust, contradicting the statement that FTLE inherits "substantial uncertainty". I noticed you cite other papers by Balasuriya. Perhaps this is where you take the view that FTLE can inherit uncertainty. However, as mentioned in my previous review, his work relies heavily on adding a stochastic component to the velocity to study uncertainty in*

*LCS. This has two problems: 1) the dynamical systems approach we are discussing (FTLE/LCS) is strictly valid for a deterministic velocity (wrong tool for the problem) and 2) a stochastic component is not representative of the uncertainty we are interested in (wrong problem). Please consider this and account for it in your paper.*

*I would also suggest changing (as this is part of what you are trying to find out):*
*In addition, velocity fields and resulting FTLE evolve rapidly …*
*To*
*In addition, velocity fields and resulting FTLE could evolve rapidly …*
*You will be better positioned to discuss the evolution of FTLE by comparing the two methods of time averaging FTLE, the one you did and the one I suggest.*

Thank you for spotting this. In the abstract we are indeed first comparing FTLE to Lagrangian trajectories, saying that both are uncertain, without much basis for it. We have done some changes in the abstract and are more careful with the wording.

*In 25 of the revised manuscript: LCS don't suggest transport barriers they identify them rigorously (at least with the theory of Haller and collaborators). FTLE do suggest transport barriers as they may or may not be LCS.*

Wording has been changed in line 25

*In 39, do you mean unsteady instead of unstable? Unstable is not a good adjective in this context. The word unstable is used for different things depending on whether you refer to LCS (unstable manifolds), any vector field as a solution to an ODE (unstable vector field), or flow instability.*

Yes, we meant to write "unsteady" here, this was an error.

*45, the method by Duran et al does not detect LCS, they detect persistent attracting structures that they call climatological LCS. However, these structures are not hyperbolic LCS. For example, they are not material lines, nor are they rigorous transport barriers, although they tend to identify persistent barriers. This comment is also relevant to other places in the papers where cLCS are referred to as LCS.*

Thank you for pointing this out. We changed the reference in line 45. The paragraph discussing cLCS (lines 337-345) has been updated to reflect the comment.

*In equation 1, the velocity also depends explicitly on the time.*

The equation has been updated.

*Equations 6 and 7 could be clearer, I would suggest dropping the $t_0$ and $t$ subscripts and superscripts (you can mention they are understood to be there after you define FTLE), using an index for the sum and express the spatial and temporal dependence of the FTLE fields (the temporal mean will not have a dependence on time, but the ensemble mean will). Also, it is unlikely N will be the same in both averages, use N and, say, M to indicate this.*

*These averages need to be clearly defined because there are at least two ways of obtaining ensemble averaged FTLE as discussed above.*

We state that the subscripts are dropped for simplicity in line 141.
Equations 6 and 7, as well as the text in lines 157-161 have been updated to reflect the reviewers comment.

*163-166 I find the wording unclear, you seem to suggest computing a time average and then computing FTLE from the time-averaged velocity? This is confusing because you can't integrate a time-averaged velocity unless you repeat the same velocity at each time as if it were a steady velocity. This averaging needs to be clarified, and I would suggest that time averaging the velocity is questionable as you want to consider some time evolution of the flow, as you correctly suggest, even if this time evolution happens after ensemble averaging the velocity, as suggested above.*

We replaced this paragraph with a paragraph about ensemble averaging velocity fields, then computing FTLE, then conducting the time-average of FTLE fields (as suggested above).

*Figure 5 caption: you don't explain what c and d are.*

Caption has been fixed.

*Figure 8, If we are comparing FLTE from different sources, why not plot them all with the same colorscale so the strengths of the FTLE can be compared? Including a colorbar and units gives you additional physical information that has value, as you do in other figures.*

Color schemes in Figure 8 (now Figure 9) have been changed, so that all subfigures now follow the same color scheme. A colorbar has also been added, and the subfigures should be more comparable now.

*This sentence in 257–259 is unclear: "depends on which members are included in the average for few members, which might affect the results as there is a possibility that any randomly selected member might deviate largely from other members."¨*

We decided to remove this sentence from the manuscript as it was confusing and didn't add much to the text.

*392 you mention Serra et al 2020 and velocity uncertainty; that paper discussed the robustness of LCS (OECS to be precise), in the context of operational use, and velocity uncertainty. This is very relevant to your topic so I suggest adding a bit of text to your discussion commenting on their findings and what that means in terms of your findings. They show (through a different method) that LCS/OECS are robust to uncertainty in the velocity. I think their discussion on this topic is in the main paper, but perhaps it is also in their supplementary information.*

Thank you for bringing this to our attention. We agree that their findings are highly relevant to our paper, and have added a paragraph about it to our discussion (lines 426-432).

---

## Author Response (AR3)

*I am glad the possibility of ensemble averaging of the flow was explored, as the conclusions from time averaging the FTLE field can be misleading in terms of assessing the persistence of an FTLE field. I believe that the result that FTLE are persistent, and that ensemble-averaging the flow is a good way to find persistent FTLE. I have a few minor suggestions and some comments on the ensemble averaging to detect FTLE. I will not require a new revision, but feel free to consider them as you see fit.*

We would like to thank the reviewer for all the constructive criticism, interpretation of results, suggestions about phrasing, and for sharing various articles relevant for our study throughout the review process, all of which have improved the quality of our manuscript.

*MINOR SUGGESTIONS*
*Lines correspond to the tracked-changes document.*
*I am not convinced by the description of FTLE in lines 131–136. I do not know about any assumption that separations would need to grow exponentially, although it is true that we seek separations that grow exponentially by taking the natural logarithm and thereby filtering out anything that did not grow exponentially. Also, FTLE does not find maximum separation rates; it gives you back all separation rates as a function of initial positions, from which you then need to search for maxima in the form of ridges.*

We have made changes regarding the assumption that separation grows exponentially (new lines 123-24) and have also reworded the initial mention of maximum separation rates (lines 126-28).

*On lines 362–363 I would change "but will not yield material convergence towards or divergence away from the FTLE ridge" for "but will not yield normal attraction towards or normal separation away from the FTLE ridge, which is what characterizes LCS"*

This is a nice suggestion which we have included in the manuscript in lines 313-315.

*497 OECS were introduced by Serra & Haller 2016*

Serra & Haller 2016 has been cited in the OECS discussion (line 432).

*ENSEMBLE AVERAGING COMMENTS*
*Lines 399–403: cLCS uses a type of ensemble averaging already, however the ensemble is not created from different models. Instead, each ensemble member is a distinct one-year simulation produced by the same model. For example, if a model simulation spans the years 1994–2012 then the first ensemble member is the simulation for 1994, the second member is 1995 and the last one is 2012. By averaging the ensemble members, a climatological velocity is obtained. This is a type of ensemble average that, very much like in your case, removes flow fluctuations that could hide the existence of persistent LCS. The second type of averaging in the cLCS process (the first one being the ensemble average that produces the climatological velocity) is the averaging of Cauchy-Green tensors, and as shown in the Supplementary Information of Duran et al 2018, this is not necessary, although it simplifies the visualization (see "Appendix C. A quantitative comparison between cLCSs and LCSs" in their Supplementary Information). This means that cLCS could be computed from any of the CG tensors without averaging them and the results would be very similar. This shows that*

*the fundamental step in finding cLCS is the ensemble averaging (computing the climatological velocity).*

*It was good to confirm that, in your study, time-averaging without ensemble averaging is misleading as it suggests that FTLE are less persistent than what they really are, and that ensemble averaging effectively identifies persistent FTLE. There are many studies that have come to similar conclusions using cLCS, an up-to-date list can be found at oceanresearch.xyz/clcs-ciam-users-worldwide/*

*In your study, we now see a consistent result but in a different setting, thus, I believe your work emphasizes the importance of ensemble averaging in the context of finding robust LCS/FTLE. I expect future research will follow this lead.*

We appreciate the reviewers comments and clarifications about the cLCS method. The paragraph pertaining to cLCS (lines 339-350) has been revised to reflect some of these comments.